# KNOWLEDGE DISTILLATION VIA FLOW MATCHING

## ABSTRACT

In this paper, we propose a novel knowledge transfer framework that introduces Rectified flow into knowledge distillation and leverages multi-step sampling strategies to achieve precision flow matching. We name this framework Knowledge Distillation via Flow Matching (FM-KD), which can be integrated with a metric-based distillation method with any form (*e.g.* vanilla KD, DKD, PKD and DIST), a meta-encoder with any available architecture (*e.g.* CNN, MLP and Swin-Transformer), and achieves significant accuracy improvement for the student. We theoretically demonstrate that the training objective of FM-KD is equivalent to minimizing the upper bound of the teacher feature map's or logit's negative log-likelihood. Besides, FM-KD can be viewed as a unique implicit ensemble method that leads to performance gains. To avoid introducing additional computational overhead in inference, we further design the lightweight FM-KD$^\Theta$. By slightly modifying the FM-KD framework, FM-KD can also be transformed into an online distillation framework OFM-KD with desirable performance gains. Through extensive experiments on CIFAR-100, ImageNet-1k, and MS-COCO datasets, we empirically validate the scalability and state-of-the-art performance of our proposed methods among relevant comparison approaches.

## 1 INTRODUCTION

Despite the remarkable achievements of deep neural networks, the dramatic increase in the number of parameters in recent years prevents their application to real-world scenarios. To solve this problem, knowledge distillation (Hinton et al., 2015) has been introduced for model compression in order to deploy lightweight models with desirable performance on mobile devices. Knowledge transfer, a critical high-level concept in the knowledge distillation framework, aims to transfer knowledge from a high-capacity teacher to a lightweight student, ensuring efficient student performance during runtime. The vast majority of existing distillation algorithms focus on exploring part components of knowledge transfer, including how to design effective and efficient meta-encoders to transform the output (*i.e.* feature or logit) of the student in the high dimensional space to match the corresponding output of the teacher (Chuanguang et al., 2021; Meng et al., 2022; Huang et al., 2022), and designing metric-based distillation methods to reduce the gap between the output of the teacher and the output of the student (Tao Huang & Xu, 2022; Zhao et al., 2022; Tung & Mori, 2019).

However, the research on the training framework for knowledge transfer has not been explored in depth (Gou et al., 2021). Most distillation methods employ a simple training paradigm for knowledge transfer. This can be simply accomplished by transforming the feature/logit of the student to get the predicted feature/logit using a meta-encoder, and then aligning the predicted feature/logit with the feature/logit of the teacher. As is well known, transferring fine-grained knowledge from the teacher to the student under a single

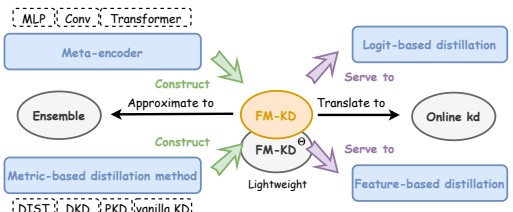

Figure 1: FM-KD is a highly scalable knowledge transfer framework.

and basic meta-encoder is challenging. An intuitive form of improvement is to increase the reliability of the output by weighted voting, or the weighted average of multiple outputs. And such an approach is frequently underemphasized in simple training paradigms. In this work, we treat the features/logits of both the teacher and the student as empirical distributions. With this approach, we can establish a flow between these two empirical distributions and then employ weighted voting

through multi-step sampling. This allows us to build a stronger knowledge transfer meta-encoder by leveraging multiple basic knowledge transfer meta-encoders, ultimately improving the student's generalization ability. Inspired by this insight and the flow matching in Rectified flow (Liu et al., 2022) that achieves accelerated convergence, we propose a novel knowledge transfer framework, Knowledge Distillation via Flow Matching (FM-KD), to amend incorrect single output from the student through multi-step sampling. In theory, FM-KD can be regarded as a unique implicit ensemble algorithm that effectively improves the performance of the student through multi-step sampling based on numerical integration. Particularly, unlike the recent work DiffKD (Huang et al., 2023) which needs to add noise and then denoise through a meta-encoder, we design an effective and desirable training objective and ensure that the gradient of the student propagates safely into the first few layers in theory, eliminating the redundant noise-adding operation during inference.

FM-KD is a versatile training paradigm for knowledge transfer with high scalability. It holds the potential to be integrated with multiple different concepts within knowledge distillation. As depicted in Fig. 1, FM-KD is comprised of a meta-encoder with any available architecture and a metric-based distillation method with any form, enabling both feature-based and logit-based distillation, and consequently enhancing the generalization ability of the student. Most importantly, it can be theoretically interpreted as an implicit ensemble algorithm. Notably, we propose a variant of FM-KD called FM-KD$^\ominus$, which avoids additional computational overhead during inference by treating FM-KD as a teacher during training and transferring the knowledge to the original network. By introducing a metric function between the predicted velocity at each time point and the numerical solution derived from the final discrete sampling, FM-KD can be transformed into an online distillation algorithm OFM-KD. Our experiments, both qualitative and quantitative, demonstrate that FM-KD and OFM-KD enhance accuracy on the image classification datasets, including CIFAR-100 and ImageNet-1k. For instance, under ResNet34-ResNet18 pair in the offline knowledge distillation scenario and ResNet18 in the online knowledge distillation scenario, FM-KD achieves the highest 72.66% and 71.56% on ImageNet-1k, respectively. Besides, FM-KD also shows effectiveness in the object detection task (MS-COCO). This highlights the potential inherent in the evolving field of study related to the design of knowledge transfer frameworks.

## 2 BACKGROUND

### 2.1 REVIEW THE KNOWLEDGE TRANSFER

Knowledge transfer plays an important role in knowledge distillation (Hinton et al., 2015), which aims to transfer the teacher's knowledge to the student, thus enhancing the performance of the student. In classical knowledge distillation algorithms, a common and simple approach (Zagoruyko & Komodakis, 2016a; Ahn et al., 2019; Tung & Mori, 2019; Xu et al., 2020; Tian et al., 2019; Cao et al., 2022; Zhao et al., 2022; Tao Huang & Xu, 2022) is to align the student's feature/logit $X^S$ with the teacher's feature/logit $X^T$ using two encoders: $g^S(\cdot)$ and $g^T(\cdot)$. This can be expressed as $\min L(g^S(X^S), g^T(X^T))$, where $L(\cdot, \cdot)$ refers to the distance metric function with any form. In some cases, $g^S(\cdot)$ and $g^T(\cdot)$ can be reduced to *the identity function*, making the supervision a direct matching between $X^S$ and $X^T$. This is widely employed in logit-based distillation.

In previous distillation algorithms, designers insisted on not altering the student architecture. As a result, $X^S$ is consistently used for forward propagation of the latter part of the network, rather than $g^S(X^S)$, following the matching criterion $\min L(g^S(X^S), g^T(X^T))$. The recently proposed DiffKD (Huang et al., 2023) replaces the traditional meta-encoder $g^S(\cdot)$ by the diffusion model for ideal encoding and reorganizes the student's architecture by replacing $X^S$ with $g^S(X^S)$ as the input for later layers. This novel approach greatly boosts the generalization ability of the student. However, DiffKD employs a large number of convolutional/linear layers in its diffusion model architecture[1], which results in a substantial increase in the inference cost. Moreover, DiffKD does not directly accomplish the translation from $X^S$ to $X^T$, but rather attempts to convert $X^S$ to a Gaussian corrupted sample first, and then convert this sample to $X^T$. This two-step conversion approach might be overly complex, hindering its widespread use. To address this problem, we introduce Rectified flow (Liu et al., 2022) to model $g^S(\cdot)$, which we will clarify later.

---

[1] https://github.com/hunto/DiffKD/blob/main/diffkd_modules.py

## 2.2 Link Rectified Flow to Knowledge Transfer

Rectified flow (Liu et al., 2022) is a simple but effective approach to model a transport map $g(\cdot)$ : $\mathbb{R}^d \to \mathbb{R}^d$ between two empirical distributions $\pi_0$ and $\pi_1$. Given the couple $(Z_0, Z_1)$ sampling from $(\pi_0, \pi_1)$, if we need to transfer empirical distribution $\pi_1$ to $\pi_0$, Rectified flow optimizes a meta-encoder $g_{v_\theta}(\cdot)$ with the parameters $v_\theta$ by solving a flow matching problem:

$$\underset{v_\theta}{\arg\min} \int_0^1 \mathbb{E}[|||(Z_1 - Z_0) - g_{v_\theta}(Z_t, t)|||]dt, \quad \text{where} \quad Z_t = tZ_1 + (1-t)Z_0. \tag{1}$$

In inference, the reverse sampling process can be achieved by solving the Ordinary Differential Equation (ODE) $\frac{d\hat{Z}_t}{dt} = -g_{v_\theta}^*(\hat{Z}_t, t)$ through the numerical integration with an initial condition $\hat{Z}_1 \sim \pi_1$ and the optimized meta-encoder $g_{v_\theta}^*(\cdot)$, which ultimately yields the synthesized data $\hat{Z}_0$ that is expected to satisfy $\hat{Z}_0 \sim \pi_0$.

Unlike the classical diffusion models (Song & Ermon, 2019; Ho et al., 2020), Rectified flow does not necessitate binding either $\pi_0$ or $\pi_1$ to a prior distribution, such as the Gaussian distribution used in diffusion models. Typically, the outputs from both the teacher and the student do not adhere to any prior distribution and are purely empirical. This flexibility in Rectified flow makes it particularly apt for knowledge transfer during knowledge distillation. It means that Rectified flow can serve as an effective tool to transfer the feature map or logit $X^S$ from the student $f^S$ to the teacher's feature map or logit $X^T$ from the teacher $f^T$.

A simple and intuitive idea is to introduce the training paradigm of Rectified flow to optimize the meta-encoder $g_{v_\theta}(\cdot)$ such that it models the transport mapping between the distribution of the student feature map or logit $\pi_S$ and the distribution of the teacher feature map or logit $\pi_T$. Given $X^S \sim \pi_S$ and $X^T \sim \pi_T$, the training objective with respect to $g_{v_\theta}(\cdot)$ can be represented simply as learning a drift force equipped to point from $X^S$ to $X^T$, which can be expressed as flow matching:

$$\underset{v_\theta}{\arg\min} \mathbb{E}_{t \sim \mathcal{U}[0,1], X^S \sim \pi_S, X^T \sim \pi_T} ||(X^S - X^T) - g_{v_\theta}(tX^S + (1-t)X^T, t)||_2^2. \tag{2}$$

However, this approach inevitably encounters three major issues: **(a)** $\hat{X}^T$ sampled from the optimized $g_{v_\theta}(\cdot)$ from a given $X^S$ can ensure that $\hat{X}^T$ follows $\pi_T$. However, it does not ensure that $X^T$ and $\hat{X}^T$ originate from the same input to guarantee the performance of $f^S$; **(b)** the optimization objective might not be feasible due to the potential inconsistencies in the shapes of $X^S$ and $X^T$ during knowledge transfer; **(c)** as $t \to 0$, the gradient norm (*i.e.* $\left|\left|\frac{\partial g_{v_\theta}(tX^S+(1-t)X^T,t)}{\partial X^S}\right|\right| = t\left|\left|\frac{\partial g_{v_\theta}(tX^S+(1-t)X^T,t)}{tX^S+(1-t)X^T}\right|\right|$) given to $v_\theta$ and to the earlier layers of the student approaches 0, making it challenging for the student to learn effectively.

## 3 Methodology

### 3.1 Knowledge Distillation via Flow Matching

We introduce FM-KD, whose overall structure is illustrated in Fig. 2. The training objective is

$$\mathcal{L}_{\text{FM-KD}} = \mathbb{E}_{(X^S, X^T, Y)} \frac{1}{N} \sum_{i=0}^{N-1} L(\mathcal{T}(Z_1 - g_{v_\theta}(Z_{1-i/N}, 1 - i/N)), X^T) + \underbrace{L(\mathcal{T}(Z_1 - g_{v_\theta}(Z_{1-i/N}, 1 - i/N)), Y)}_{\text{match the ground truth label (optional)}},$$

the sampling process: $Z_{1-i/N} = Z_{1-(i-1)/N} - g_{v_\theta}(Z_{1-(i-1)/N}, 1 - (i-1)/N)/N, \quad s.t. \quad i \geq 1,$

$$\tag{3}$$

where $L(\cdot, \cdot)$ and $Y$ is the metric-based distillation method (*i.e.* the loss function) and the ground truth label, respectively. The initial state of the sampling is $Z_1 = X^S$. We define $N$ and $K$ as the number of sampling steps during training and inference, respectively. In our work, different values of $K$ are implemented using skip-step sampling of DDIM (Song et al., 2023a). The pseudo code of FM-KD can be found in Appendix A.

FM-KD addresses the issues discussed in the preceding section. Specifically, compared with the irrational Eq. 2, $\mathcal{L}_{\text{FM-KD}}$ has undergone a series of improvements: **(1)** under general conditions, $X_S$ and $X_T$ are non-independent and paired one-to-one; **(2)** since the shape of the output of $g_{v_\theta}(\cdot)$ is

always guaranteed to be the same as $X^S$ but different from $X^T$, we add the shape transformation function $\mathcal{T}(\cdot)$. This shape alignment ensures the calculation of $\mathcal{L}_{\text{FM-KD}}$; **(3)** a serial loss calculation is used to avoid gradient vanishing in the student. In our experiments, it is guaranteed that $N$ does not exceed 8 (default $N$ as 8) thereby avoiding a significant increase in computational cost. For the third improvement, the GFLOPs and the parameters of the meta-encoder remain relatively small. More details can be found in Sec. 4.3. It is important to clarify that not only the training of FM-KD needs to be performed by serial, but the inference also relies on multi-step sampling.

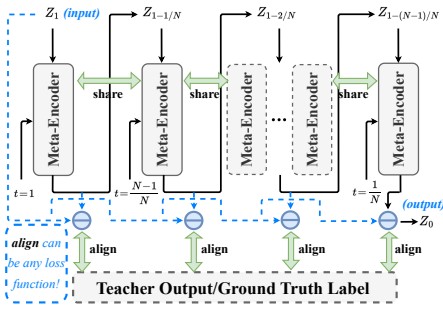

Figure 2: The overall structure of FM-KD.

Intuitively, FM-KD is an interesting "time-for-accuracy" algorithm, which in some ways makes a trade-off between time cost and student performance even in inference.

**Theorem 3.1.** *(Proof in Appendix B) Optimizing $\mathcal{L}_{\text{FM-KD}}$ not only ensures that the gradient of the student can be back-propagated to its earlier layers, but also provides an equivalence to the upper bound of the negative log-likelihood of $X^T$.*

To determine the feasibility and reasonableness of the revised optimization objective $\mathcal{L}_{\text{FM-KD}}$, we furnish a theoretical proof in Theorem 3.1 to ascertain that it is closely equivalence to the minimization of the upper bound of the negative log-likelihood of $X^T$. This suffices to corroborate the effectiveness of the training paradigm. Additionally, the deterministic sampling process in FM-KD facilitates the transfer of $X^S$ to $X^T$ when $\mathcal{L}_{\text{FM-KD}} \to 0$ (*i.e.* the student converges).

## 3.2 SERVE TO FEATURE-/LOGIT-BASED DISTILLATION

By simplistically integrating FM-KD into the standard distillation framework, it can serve to the majority of feature-/logit-based distillation algorithms. This introduction is straightforward; it involves replacing the loss function in FM-KD with suitable metric-based distillation approaches. Practically, FM-KD is strategically placed between different layers of the student to accomplish knowledge transfer. We give an example in Fig. 3. For feature-based distillation, FM-KD is inserted between the intermediate layers of the student, typically before the downsampling layer. This insertion does not alter the rest of the student architecture. Furthermore, for logit-based distillation, FM-KD replaces the original pooling layer,

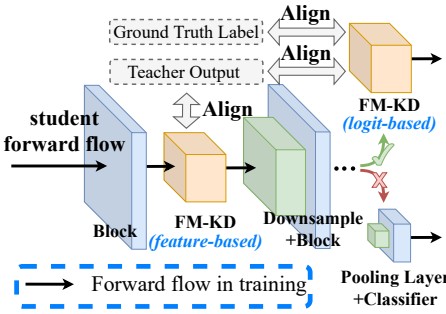

Figure 3: An example of FM-KD usage.

linear classification layer, or even the penultimate one or two layers (*e.g.* convolution, activation and normalization layers), to achieve logit-level matching. In our experiments, the unique replacement of the extra penultimate one or two layers is only used for the student on CIFAR-100 is MobileNetV2. Besides, as shown in Eq. 3, FM-KD can optionally add a new loss function by substituting the ground truth label for $X^T$, enabling consistency with the classical logit-based distillation paradigm.

For complex distillation algorithms with learnable encoders, such as MasKD (Yang et al., 2022), we can denote the entire algorithm as a loss function. Hence, it is plausible to replace $L(\cdot, \cdot)$ in FM-KD with these algorithms to enable "serve to feature-/logit-based distillation". In this study, we focus on simple yet effective metric-based distillation methods, including vanilla KD, DKD, PKD, and DIST. The adaptation of more complex distillation algorithms is earmarked for future work, which will help further ascertain the robust applicability of FM-KD.

## 3.3 HIGH LEVEL OF APPLICABILITY

FM-KD is not tightly bound to the meta-encoder and the metric-based distillation method, so it can be amalgamated with any of them, thereby leveraging their benefits within this unique framework. The architecture of the meta-encoder and the form of the metric-based distillation method employed for flow matching can be selected in any viable form. In this study, we use MLP, CNN (He et al., 2016), and Swin-Transformer (Liu et al., 2021) as alternatives for the meta-encoder, and vanilla KD, DKD (Zhao et al., 2022), PKD (Cao et al., 2022), and DIST (Tao Huang & Xu, 2022) as alternatives for the loss function.

### 3.4 APPROXIMATE TO ENSEMBLE

We attribute the ability of FM-KD to efficiently realize knowledge transfer to its multi-step sampling enabled by numerical integration. Although in the training and inference phases of distillation, we control $N$ ($\leq 8$) to be small enough that it no longer satisfies the form of a continuous ODE probability flow. However, as shown in Eq. 3, Euler's method can be rewritten as averaging multiple time-step outputs, which intuitively approximates ensemble approaches. For completeness, we provide in-depth theoretical support for this argument through the perspective of error analysis in Proposition 3.2.

**Proposition 3.2.** *(Proof in Appendix C) FM-KD can be considered a unique implicit ensemble algorithm. The number of outputs used for ensemble is equivalent to the number of samplings.*

As is well-known, some past methods (Lu et al., 2022; Song et al., 2023b) for error analysis in the sampling process of the diffusion model use absolute error bound, which achieves recursion and thus scaling of the accumulated error value. We discard the constraint on the absolute value and employ recursion and Taylor expansion in the derivation of Proposition 3.2. As a result, we obtain the interesting conclusion that the truncation error, which is supposed to be progressively scaled, makes the sampling process of the FM-KD a unique implicit ensemble approach in this proposition.

### 3.5 LIGHTWEIGHT FM-KD$^\Theta$ WITHOUT ADDITIONAL INFERENCE BURDEN

FM-KD introduces additional overhead during inference. To facilitate efficient deployment, we propose a streamlined variant of FM-KD for logit-based distillation, referred to as FM-KD$^\Theta$. This variant enhances the process by distilling $Z_0$ from FM-KD into the existing classification head (*i.e.* the original student's classification head) $\mathcal{T}_{\text{vanilla}}(\cdot)$, ensuring no extra inference cost. Essentially, this is a concept of progressive distillation, which enhances student performance by effectively reducing the gap between the teacher and the student. During training, we reformulate the loss function to accommodate this integration:

$$\mathcal{L}_{\text{FM-KD}^\Theta} = \mathbb{E}_{(X^S, X^T, Y)} L(\mathcal{T}_{\text{vanilla}}(X^S), \mathcal{T}(Z_0)) + \alpha^\Theta L(\mathcal{T}_{\text{vanilla}}(X^S), Y) + \mathcal{L}_{\text{FM-KD}}, \tag{4}$$

where $\alpha^\Theta$ refers to the balance weight. In inference, we can directly utilize $\mathcal{T}_{\text{vanilla}}(\cdot)$ to achieve prediction without going through $g_{v_\theta}(\cdot)$ and $\mathcal{T}(\cdot)$ to increase the sampling burden.

### 3.6 TRANSLATE TO ONLINE KNOWLEDGE DISTILLATION

Numerous Online Knowledge Distillation (Online KD) algorithms essentially integrate the outputs of multiple branches, thus avoiding asynchronous updating of gradients and ultimately improving the generalization ability of the student. FM-KD and Online KD have different approaches but equally satisfactory results, which provides the feasibility for FM-KD to be converted to Online KD. In comparison to Offline Knowledge Distillation (Offline KD), Online KD doesn't use an explicit teacher; instead, the teacher is represented by a weighted average of branches in the student. Similarly, we can achieve the goal "translate to Online KD" by simply replacing $X^T$ in Eq. 3 with the final result after sampling with Euler's method. In detail, we first obtain the sampling result $Z_0$ by continuously calling Euler's method $Z_{1-i/N} = Z_{1-(i-1)/N} - g_{v_\theta}(Z_{1-(i-1)/N}, 1-(i-1)/N)/N$. Finally, we retain the portion of FM-KD that matches the ground truth label and add the Online KD loss to it

$$\mathcal{L}_{\text{OFM-KD}} = \mathbb{E}_{(X^S, Y)} \frac{1}{N} \sum_{i=0}^{N-1} \underbrace{L(Z_1 - g_{v_\theta}(Z_{1-i/N}, 1-i/N), Z_0)}_{\text{the Online KD loss}} + \underbrace{L(\mathcal{T}(Z_1 - g_{v_\theta}(Z_{1-i/N}, 1-i/N)), Y)}_{\text{match the ground truth label}}. \tag{5}$$

The variant $\mathcal{L}_{\text{OFM-KD}}$ can be empirically understood as a novel Online KD algorithm OFM-KD. Compared with traditional Online KD algorithms including ONE (Chen et al., 2020b), KDCL (Guo et al., 2020) and AHBF-OKD (Gong et al., 2023), OFM-KD has some unique characteristics, including the meta-encoder shares parameters at different time points, whereas traditional Online KD algorithms do not shares parameters at different branches. Besides, the input of the meta-encoder in OFM-KD is different at different time points, and as $t \to 0$, the input contains more target information. In contrast, the traditional Online KD has the same input for each branch. This means that OFM-KD achieves ensemble through various inputs instead of unshared parameters.

| Teacher | ResNet56 | WRN-40-2 | WRN-40-2 | ResNet32×4 | VGG13 | VGG13 | WRN-40-2 |
| Student | ResNet20 | WRN-16-2 | WRN-40-1 | ResNet8×4 | VGG8 | MobileNetV2 | ShuffleNetV1 |
|---|---|---|---|---|---|---|---|
| Teacher | 73.24 | 75.61 | 75.61 | 79.42 | 74.64 | 75.61 | 75.61 |
| *Student* | *69.06* | *73.26* | *71.98* | *72.50* | *70.36* | *64.60* | *70.50* |
| ATKD | 70.55 | 74.08 | 72.77 | 73.44 | 71.43 | 59.40 | 72.73 |
| SPKD | 69.67 | 73.83 | 72.43 | 72.94 | 72.68 | 66.30 | 74.52 |
| CRD | 71.16 | 75.48 | 74.14 | 75.51 | 73.94 | 69.73 | 76.05 |
| vanilla KD | 70.66 | 74.92 | 73.54 | 73.33 | 72.98 | 67.37 | 74.83 |
| DKD | 71.97 | 76.24 | 74.81 | 76.32 | 74.68 | 69.73 | 76.70 |
| DIST | 71.26 | 75.29 | 74.42 | 75.79 | 73.11 | 68.48 | 75.23 |
| FM-KD$^\ominus$ | 72.20 | 75.98 | 74.99 | 76.52 | 74.82 | 69.90 | 77.19 |
| DiffKD | 71.92 | 76.13 | 74.09 | 76.31 | - | - | - |
| FM-KD ($K$=1) | 74.28 | 77.14 | 75.88 | 76.74 | 75.21 | 69.68 | 76.34 |
| FM-KD ($K$=2) | 74.09 | 76.58 | 74.52 | 74.98 | 74.86 | 69.52 | 75.55 |
| FM-KD ($K$=4) | **75.12** | 77.69 | **76.24** | 77.49 | 75.42 | **69.94** | 76.95 |
| FM-KD ($K$=8) | 74.97 | **77.84** | 76.09 | **77.71** | **75.46** | **69.94** | **77.21** |

Table 1: Results of different Offline KD methods on CIFAR-100. Among them, ATKD, SPKD, CRD and DiffKD belong to feature-based distillation, while vanilla KD, DKD and DIST belong to logit-based distillation.

| T-S Pair | Accuracy | Tea. | *Stu.* | vanilla KD | ReviewKD | DKD | DIST | FM-KD$^\ominus$ | DiffKD | FM-KD ($K$=1) | FM-KD ($K$=2) | FM-KD ($K$=4) | FM-KD ($K$=8) |
|---|---|---|---|---|---|---|---|---|---|---|---|---|---|
| R34-R18 | Top-1 | 73.31 | *69.75* | 70.66 | 71.61 | 71.70 | 72.07 | 72.14 | 72.49 | 72.49 | 72.86 | 73.08 | **73.17** |
| | Top-5 | 91.42 | *89.08* | 89.88 | 90.51 | 90.41 | 90.42 | 90.44 | 90.71 | 90.83 | 91.00 | 91.12 | **91.18** |
| R50-MBV1 | Top-1 | 76.16 | *70.13* | 70.68 | 72.56 | 72.05 | 73.24 | 73.29 | 73.78 | 73.61 | 74.01 | 74.20 | **74.22** |
| | Top-5 | 92.86 | *89.49* | 90.30 | 91.00 | 91.05 | 91.12 | 91.15 | 91.48 | 91.36 | 91.71 | **91.84** | 91.81 |

Table 2: Results of different Offline KD methods on ImageNet-1k. "R34-R18" and "R50-MBV1" refer to "ResNet34-ResNet18 pair" and "ResNet50-MobileNetV1 pair", respectively.

| Method | Schedule | mAP | AP$_{50}$ | AP$_{75}$ | AP$_S$ | AP$_M$ | AP$_L$ |
|---|---|---|---|---|---|---|---|
| Mask RCNN-Swin (T) | 3×+ms | 48.2 | 69.8 | 52.8 | 32.1 | 51.8 | 62.7 |
| *Retina-Res50 (S)* | *2×* | *37.4* | *56.7* | *39.6* | *20.0* | *40.7* | *49.7* |
| PKD | 2× | 41.3 (+3.9) | 60.5 | **44.1** | **23.0** | 45.3 | **55.9** |
| FM-KD ($K$=1) | 2× | **41.4 (+4.0)** | **60.6** | 44.0 | 22.5 | **45.6** | 55.7 |
| FM-KD ($K$=4) | 2× | **41.4 (+4.0)** | **60.6** | **44.1** | 22.5 | **45.6** | 55.7 |
| FasterRCNN-Res101 (T) | 2× | 39.8 | 60.1 | 43.3 | 22.5 | 43.6 | 52.8 |
| *FasterRCNN-Res50 (S)* | *2×* | *38.4* | *59.0* | *42.0* | *21.5* | *42.1* | *50.3* |
| GID | 2× | 40.2 (+1.8) | 60.7 | 43.8 | 22.7 | 44.0 | 53.2 |
| FRS | 2× | 40.4 (+2.0) | **60.8** | 44.0 | **23.2** | 44.4 | 53.1 |
| FGD | 2× | 40.4 (+2.0) | 60.7 | **44.3** | 22.8 | 44.5 | **53.5** |
| PKD | 2× | 40.3 (+1.9) | **60.8** | 44.0 | 22.9 | 44.5 | 53.1 |
| FM-KD ($K$=1) | 2× | 40.4 (+2.0) | 60.7 | 44.1 | 22.9 | **44.8** | 52.8 |
| FM-KD ($K$=4) | 2× | **40.5 (+2.1)** | 60.7 | 44.2 | 22.9 | **44.8** | 52.9 |
| FCOS-Res101 (T) | 2×+ms | 41.2 | 60.4 | 44.2 | 24.7 | 45.3 | 52.7 |
| *Retina-Res50 (S)* | *1×* | *37.4* | *56.7* | *39.6* | *20.0* | *40.7* | *49.7* |
| PKD | 1× | 40.3 (+2.9) | 59.6 | 43.0 | 22.2 | 44.9 | **53.7** |
| FM-KD ($K$=1) | 1× | **40.5 (+3.1)** | **59.9** | **43.6** | **22.5** | 45.0 | 53.5 |
| FM-KD ($K$=4) | 1× | **40.5 (+3.1)** | 59.8 | **43.6** | **22.5** | 45.0 | **53.7** |

Table 3: Results of FM-KD with different detection frameworks on MS-COCO. "T" and "S" mean the "teacher" and "student" detector, respectively.

## 4 EXPERIMENT

We perform comparison and ablation experiments on CIFAR-100, ImageNet-1k and MS-COCO. The implementation details of FM-KD, FM-KD$^\ominus$, and OFM-KD can be found in Appendix I. Note that all normalization layers in the meta-encoder are not BatchNorm, because their inputs are various at different time points, so the statistics of the mean and variance will encounter difficulties, thereby causing training collapse. Moreover, we introduce a strategy named Pair Decoupling (PD), which is controlled by the hyperparameter dirac ratio $\beta_d$, applied to shuffle part of the sample pairs in a batch. This approach is particularly effective for feature-based distillation in image classification tasks, and its detailed description and specific implementation can be found in Appendix D and A, respectively. The impact of the normalization layer selection in the meta-encoder, where stages used for distillation in the feature-based scenario, and ideal configuration of dirac ratio $\beta_d$ can be found in the additional ablation experiments in Appendix E. By default, we set $\beta_d$ as 0.25 and use the 1st and 2nd last stages for feature-based distillation in image classification tasks.

| Architecture | ResNet32 | ResNet110 | VGG16 | DenseNet40-2 | MobileNetV2 |
|---|---|---|---|---|---|
| *Student* | *71.28* | *76.21* | *74.32* | *71.03* | *59.79* |
| CL | 72.33 | 78.83 | 74.33 | 71.45 | 60.63 |
| ONE | 72.45 | 78.44 | 74.38 | 71.39 | 60.84 |
| FFSD-C | 74.50 | 78.83 | 74.89 | 71.74 | 61.88 |
| ABHF-OKD | **74.81** | 79.04 | 75.08 | 72.12 | 62.23 |
| OFM-KD ($K$=1) | 72.86 | 79.49 | 75.07 | 73.12 | 63.62 |
| OFM-KD ($K$=2) | 73.02 | **79.50** | **75.10** | 73.34 | **63.67** |
| OFM-KD ($K$=4) | 73.10 | 79.45 | 75.09 | **73.40** | 63.63 |
| OFM-KD ($K$=8) | 73.07 | 79.47 | 75.06 | 73.39 | 63.61 |

Table 4: Results of different Online KD methods on CIFAR-100. The metric is the Top-1 accuracy.

| Architecture | *Student* | ONE | OKDDip | FFSD-C | ABHF-OKD | OFM-KD ($K$=1) | OFM-KD ($K$=2) | OFM-KD ($K$=4) | OFM-KD ($K$=8) |
|---|---|---|---|---|---|---|---|---|---|
| ResNet18 | *69.75* | 70.55 | 70.63 | 70.15 | 70.72 | 71.38 | 71.52 | **71.56** | **71.56** |
| ResNet34 | *73.24* | 74.10 | 74.40 | 74.20 | **74.53** | 74.16 | 74.20 | 74.20 | 74.20 |

Table 5: Results of different Online KD methods on ImageNet-1k. The metric is the Top-1 accuracy.

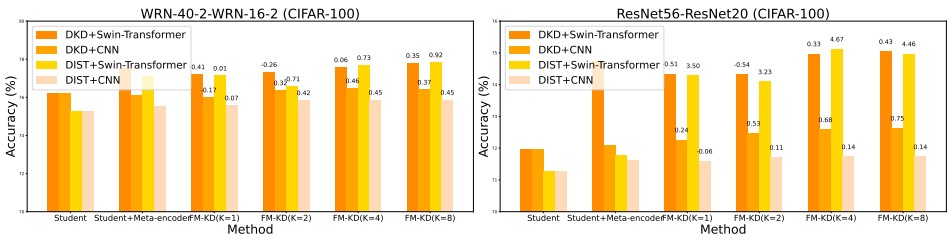

Figure 4: Results of experiments on the ensemble capabilities of FM-KD on CIFAR-100. The numbers on the bars represent their performance gains compared to Student+Meta-encoder.

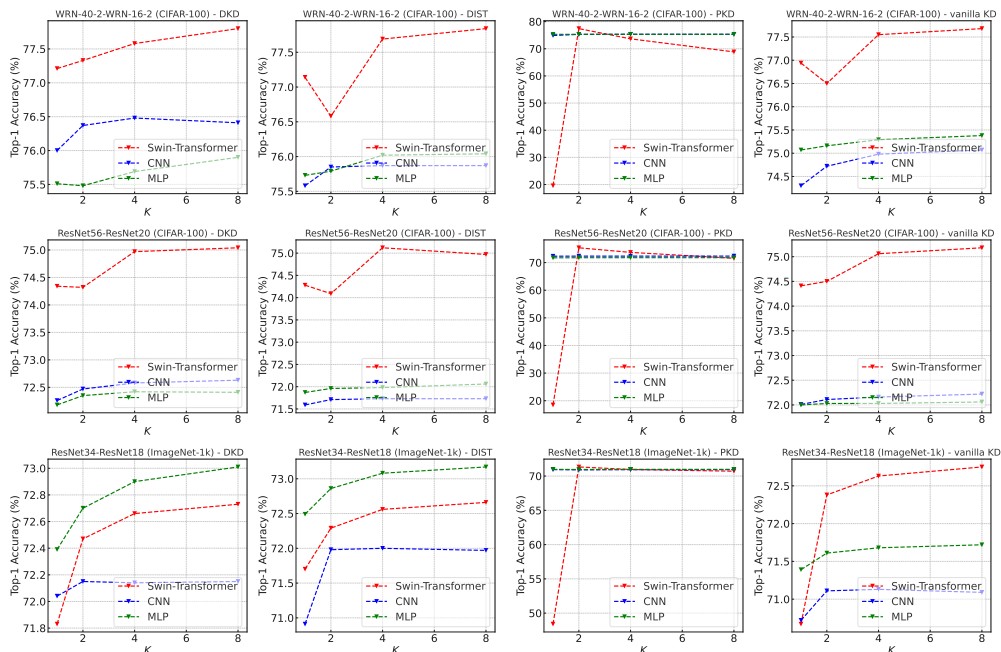

Figure 5: Ablation results about the loss function and the meta-encoder on CIFAR-100 and ImageNet-1k.

## 4.1 IMAGE CLASSIFICATION COMPARISON

**Offline Knowledge Distillation.** On CIFAR-100, we conduct experiments on teacher-student pairs including ResNet56-ResNet20, WRN-40-2-WRN-16-2 (Zagoruyko & Komodakis, 2016b), WRN-40-2-WRN-40-1, ResNet32×4-ResNet8×4, VGG13-VGG8 (Szegedy et al., 2015), VGG13-MobileNetV2 (Sandler et al., 2018) and WRN-40-2-ShuffleNetV1 (Zhang et al., 2018) pairs. We

compare FM-KD with state-of-the-art methods including ATKD (Zagoruyko & Komodakis, 2016a), SPKD, CRD, DiffKD, vanilla KD, DKD and DIST, and present the results in Table 1. As shown in Table 1, FM-KD significantly outperforms prior KD methods with all pairs. Note that FM-KD improves the student performance on ResNet56-ResNet20, WRN-40-2-WRN-16-2, WRN-40-2-WRN-40-1 and VGG13-VGG8 pairs by 3.15%, 1.60%, 1.43% and 0.64%, respectively, compared with the best prior methods. Moreover, our lightweight variant FM-KD$^\Theta$, which without additional computational cost in inference, achieves state-of-the-art performance across a wide range of teacher-student pairs. On ImageNet-1k, FM-KD treats DIST as its $L(\cdot, \cdot)$ (*w.r.t.* baseline). Compared with DIST, FM-KD exceeds DIST on ResNet34-ResNet18 and ResNet50-MobileNetV1 by 1.10% and 0.98%, respectively. In particular, compared with DiffKD, an algorithm with some similarity to FM-KD, FM-KD outperforms DiffKD on ResNet34-ResNet18 and ResNet50-MobileNetV1 by a margin of 0.68% and 0.44%, respectively. However, it should be noted that DiffKD introduces 11 additional convolutional layers in its encoder (considering its mentioned Diffusion Model and Noise Adapter), while in contrast, FM-KD employs 2-layer MLP with only 4 linear layers as its meta-encoder. Furthermore, the lightweight FM-KD$^\Theta$ outperforms all algorithms with no additional computational overhead in inference, validating its effectiveness and applicability. Finally, note that more detailed results about the stronger strategies and the stronger teacher comparison as well as the visualization of sampling trajectory can be found in Appendix G and H, respectively.

**Online Knowledge Distillation.** We present the results of the comparison between OFM-KD and prior state-of-the-art approaches CL (Song & Chai, 2018), ONE (Chen et al., 2020b), OKD-Dip (Chen et al., 2020a), FFSD-C (Li et al., 2022) and ABHF-OKD (Gong et al., 2023) in Table 4 and 5. Among them, Table 4 illustrates the experimental results on CIFAR-100. We can observe OFM-KD beats all comparison methods on ResNet110, VGG16, DenseNet40-2, and MobileNetV2. For the results on ImageNet-1k on Table 5, OFM-KD outperforms other methods on ResNet18, albeit lagging behind the optimal ABHF-OKD by a marginal 0.33% on ResNet34. Importantly, regarding both ResNet18 and ResNet34, OFM-KD necessitates merely two Number of Function Evaluations (NFEs) to attain the best results. This indicates that OFM-KD corresponds to Online KD, which is the aggregated outcome of two branches sharing parameters. Hence, this compellingly substantiates that OFM-KD is a potent Online KD algorithm.

## 4.2 OBJECT DETECTION COMPARISON

The experimental results of object detection are presented in Table 3, where Mask RCNN-Swin-RetinaNet-Res50 pair represents the case of being distilled from a strong teacher, FasterRCNN-Res101-FasterRCNN-Res50 pair represents the homogeneous teacher-student pair, and FCOS-Res101-Retina-Res50 pair represents the heterogeneous teacher-student pair. We observe that FM-KD, which applies PKD as its loss function, shows improvement to some extent compared to the baseline FKD and achieves state-of-the-art performance across all teacher-student pairs. Note that knowledge transfer in object detection is facilitated by the high similarity between the feature maps of the student and the teacher. Consequently, the student's mAP remains consistent for both $K=4$ and $K=8$, so we do not present results for $K=8$.

## 4.3 ABLATION STUDIES

**The Number of Sampling Steps $K$ in Inference.** We can also call $K$ as NFEs, an important metric affecting the GPU latency during inference. Both FM-KD and OFM-KD have a similarity form to the diffusion models family (*e.g.* VE-SDE, VP-SDE, EDM (Karras et al., 2022)) and INN (Solodskikh et al., 2023)), in that after obtaining the training weights, the NFEs can be modified at the time of inference to trade-off effective and efficiency. We can see from Table 1, 4, 2, 5 and Fig. 5 that increasing $K$ will improve the student performance, but in general $K = 2$ will achieve quite satisfactory results. Note that the combination of PKD and Swin-Transformer on ImageNet-1k in Fig. 5 has a large difference in the results achieved by the different NFEs, but the best results are superior to the combination of PKD and MLP/CNN.

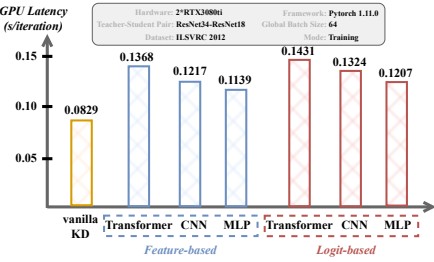

Figure 6: GPU latency comparison between FM-KD and vanilla KD.

This might be because Swin-Transformer does not have inductive bias (Park & Kim, 2021) and the specificity that PKD is a feature-based distillation method.

**The Training Computational Cost Analysis.** As illustrated in Fig. 6, applying a serial loss calculation $\mathcal{L}_{\text{FM-KD}}$ does not introduce excessive GPU latency during training. Even the most computationally demanding meta-encoder Transformer has less than double the GPU latency compared to vanilla KD. Compared with DPK (Zong et al., 2023), which uses 6 encoders and 4 decoders, resulting in a GPU training latency more than 6 times that of FitNet (Romero et al., 2014), the computational cost introduced by FM-KD is not huge.

**The Effectiveness of Optimization Objective.** We present the validity of the FM-KD optimization objective in Fig. 4 to prevent misinterpretations due to the properties of the meta-encoder and the loss function themselves. As $K$ rises, we can observe that the performance improvement becomes increasingly clear. Note that a increase of 4.67% is specifically produced by the DIST+Swin-Transformer on the ResNet56-ResNet20 pair. This demonstrates that the characteristic of FM-KD – implicit ensemble – does result in performance gains.

**The Ablation about the Loss Function and Meta-Encoder.** The outcomes of this ablation study are summarized in Fig. 5. For the meta-encoder, it is demonstrated that the Swin-Transformer yields the most favorable results when combined with any loss function on CIFAR-100. Conversely, on ImageNet-1k, the amalgamation of MLP with DKD, DIST and PKD demonstrates superior performance. Moreover, for the loss function, DIST and DKD exhibit comparable and enhanced performance relative to PKD and vanilla KD across all student-teacher pairs.

## 5 RELATED WORK

**Knowledge distillation.** A technique for model compression, effectively enhances the performance of lightweight models. The main strategies of knowledge distillation are categorized into three: feature-based (Zagoruyko & Komodakis, 2016a; Tung & Mori, 2019; Tian et al., 2019; Li et al., 2022; Zhang & Ma, 2020; Huang et al., 2023), logit-based (Tao Huang & Xu, 2022; Zhao et al., 2022; Hinton et al., 2015; Shen & Xing, 2022), and data-based distillations (Wang et al., 2022; Shao et al.). Regardless of the approach, the knowledge transfer framework plays an important role in it. Thus, this paper aims to design a more desirable knowledge transfer framework that can serve both feature-based distillation as well as logit-based distillation.

**Continuous Network Representation.** There are a number of architectures belonging to continuous network representation, such as RNN (Williams & Zipser, 1989), LSTM (S & J, 1997), Neural ODE (Chen et al., 2018), GflowNet family (Bengio et al., 2021; Zhang et al., 2022), diffusion model family (Song et al., 2023c; Karras et al., 2022; Ho et al., 2020; Song et al., 2023a), INN (Solodskikh et al., 2023) and DiffKD (Huang et al., 2023). Compared with these methods, FM-KD is an efficient and effective continuous network representation with training stability applied to knowledge transfer. More discussion can be found in Appendix F.

## 6 CONCLUSION

We have proposed a highly scalable framework FM-KD, and its lightweight variant FM-KD$^{\Theta}$, for knowledge transfer in knowledge distillation. Additionally, we introduced its variant, OFM-KD, for the Online KD paradigm. The design flexibility of both FM-KD and OFM-KD allows them to be formulated utilizing a loss function of any form and a meta-encoder with any available architecture, making them adaptable for distillation processes focused both on features and logits. Theoretically, we have proven that the optimization objective of FM-KD is equivalent to minimizing the upper bound of the negative log-likelihood of the target (*e.g.* the teacher's output). Moreover, we link the characteristics of multi-step sampling with FM-KD and OFM-KD, ensuring that they empower the student with remarkable generalization capabilities. In future work, we aim to further explore the design space of FM-KD and extend its application to a broader scope of downstream tasks.

## 7 LIMITATION

FM-KD demonstrates improved generalization capabilities relative to conventional KD methods, yet it incurs a higher computational burden during inference. Moreover, FM-KD's effectiveness in object detection is not as pronounced as in image classification. This discrepancy stems from the fact that, in image classification, flow matching with the teacher at the logit level often yields performance akin to the teacher's. In contrast, in object detection, flow matching with the teacher at the FPN (Feature Pyramid Network) level does not directly translate to enhanced performance in the ultimate metric, mAP.

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

# A  PSEUDO CODE OF FM-KD

For ease of understanding, we show the pseudo code of FM-KD in the Offline KD scenario. The implementation of OFM-KD only needs to modify the optimization objective as described in our main paper.

---

**Algorithm 1** Pseudo code of FM-KD in a PyTorch-like style.

---

```python
import torch
import torch.nn as nn
import torch.nn.functional as F

class FlowMatchingModule(nn.Module):
    def __init__(self,...):
        super().__init__()
        self.meta_encoder:nn.Module = (...)
        self.metric_based_loss_function:nn.Module = (...)
        self.time_embed:nn.Module = nn.Linear(...)
        self.training_sampling:int = (...)   # the number of sampling steps during training
        self.shape_transformation_function:nn.Module = (...)
        self.dirac_ratio:float = (...)   # hyperparameter β_d, which belongs to [0,1]
        self.weight:float = (...)

    def forward(self, s_f, t_f=None, target=None, inference_sampling=1):
        # s_f: the feature/logit of the student
        # t_f: the feature/logit of the teacher
        # target: the logit-based ground truth label, only used for logit-based distillation
        # inference_sampling: the number of sampling steps during inference

        all_p_t_f = []
        if self.training:
            # Shuffle one-to-one teacher-student feature/logit pair
            if t_f is not None:
                l = int(self.dirac_ratio * t_f.shape[0])
                t_f[l:][torch.randperm(t_f.shape[0] - l)] = t_f[l:].clone()
            loss, x = 0., s_f
            indices = reversed(range(1, self.training_sampling + 1))
            # Calculate the FM- KD loss
            for i in indices:
                t = torch.ones(s_f.shape[0]) * i / self.training_sampling
                embed_t = self.time_embed(t)
                embed_x = x + embed_t
                velocity = self.meta_encoder(embed_x)
                x = x - velocity / self.training_sampling
                p_t_f = self.shape_transformation_function(s_f - velocity)
                all_p_t_f.append(p_t_f)
                loss += self.metric_based_loss_function(p_t_f, t_f)
                if target is not None:
                    loss += F.cross_entropy(p_t_f, target)
            loss *= (self.weight / self.training_sampling)
            return loss, torch.stack(all_p_t_f, 0).mean(0)
        else:
            x = s_f
            indices = reversed(range(1, inference_sampling + 1))
            for i in indices:
                t = torch.ones(s_f.shape[0]) * i / inference_sampling
                embed_t = self.time_embed(t)
                embed_x = x + embed_t
                velocity = self.meta_encoder(embed_x)
                x = x - velocity / inference_sampling
                all_p_t_f.append(self.shape_transformation_function(s_f - velocity))
            return 0., torch.stack(all_p_t_f, 0).mean(0)
```

---

# B  THEORETICAL GUARANTEES OF FM-KD

FM-KD proposes a novel training paradigm in order to ensure that the gradient of the student can successfully perform backpropagation:

$$\mathcal{L}_{\text{FM-KD}} = \mathbb{E}_{(X^S, X^T)} \frac{1}{N} \sum_{i=1}^{N} ||\mathcal{T}(Z_1 - g_{v_\theta}(Z_{1-(i-1)/N}, 1 - (i-1)/N)) - X^T||_2^2, \tag{6}$$

where $Z_1 = X^S$. Here we assume the loss function is $\ell_2$-norm. Broadly speaking, the loss function used by FM-KD only needs to ensure that it can achieve the effect of minimizing the difference in distributions similar to Kullback-Leibler Divergence. When $i \geq 1$, $Z_{1-i/N} = Z_{1-(i-1)/N} - g_{v_\theta}(Z_{1-(i-1)/N}, 1 - (i-1)/N)/N$. $\mathcal{T}(\cdot)$ is used for shape alignment to ensure the calculation of $\ell_2$-norm. Let us define $q(\cdot|\cdot)$ as the predefined conditional probability, and $p_{v_\theta}(\cdot|\cdot)$ as the predicted

conditional probability. We know the training objective of the classical diffusion probability model can be performed by minimizing the upper bound on negative log-likelihood:

$$-\log p_{v_\theta}(Z_0) \leq \mathbb{E}_{q(Z_{1/N:1}|Z_0)}\left[\log \frac{q(Z_1|Z_0)}{p_{v_\theta}(Z_1)p_{v_\theta}(Z_0|Z_{1/N})} + \sum_{i=1}^{N}\log\frac{q(Z_{(i-1)/N}|Z_{i/N},Z_0)}{p_{v_\theta}(Z_{(i-1)/N}|Z_{i/N})}\right]. \quad (7)$$

We can rewritten it as

$$-\log p_{v_\theta}(Z_0) \leq \mathbb{E}_{q(Z_{1/N:1}|Z_0)}\left[\log \frac{q(Z_1|Z_0)}{p_{v_\theta}(Z_1)p_{v_\theta}(Z_0|Z_{1/N})} + \sum_{i=1}^{N}\log\frac{q(Z_{(i-1)/N}|Z_{i/N},Z_0)}{p_{v_\theta}(Z_{(i-1)/N}|Z_{i/N})}\right]$$

$$= \mathbb{E}_{q(Z_{1/N:1}|Z_0)}\left[\log \frac{q(Z_1|Z_0)}{p_{v_\theta}(Z_1)p_{v_\theta}(Z_0|Z_{1/N})}\right] + \sum_{i=1}^{N}\mathbb{E}_{q(Z_{i/N}|Z_0)}\mathbb{E}_{q(Z_{(i-1)/N}|Z_{i/N},Z_0)}\left[\log\frac{q(Z_{(i-1)/N}|Z_{i/N},Z_0)}{p_{v_\theta}(Z_{(i-1)/N}|Z_{i/N})}\right]$$

$$= \mathbb{E}_{q(Z_{1/N:1}|Z_0)}\left[\log \frac{q(Z_1|Z_0)}{p_{v_\theta}(Z_1)p_{v_\theta}(Z_0|Z_{1/N})}\right] + \sum_{i=1}^{N}\mathbb{E}_{\hat{Z}_{i/N}\sim\int p_{v_\theta}(Z_{i/N}|Z_1)q(Z_1|Z_0)dZ_1}$$

$$\left[D_{\mathrm{KL}}(q(Z_{(i-1)/N}|Z_{i/N},Z_0)||p_{v_\theta}(Z_{(i-1)/N}|\hat{Z}_{i/N}))\right], \quad s.t. \quad \mathrm{Law}(Z_{i/N}) \cong \mathrm{Law}(\hat{Z}_{i/N})$$

$$\approx \mathbb{E}_{q(Z_{1/N:1}|Z_0)}\left[\log \frac{q(Z_1|Z_0)}{p_{v_\theta}(Z_1)p_{v_\theta}(Z_0|Z_{1/N})}\right] + \sum_{i=1}^{N}\mathbb{E}_{\hat{Z}_{i/N}\sim\int p_{v_\theta}(Z_{i/N}|Z_1)q(Z_1|Z_0)dZ_1}$$

$$\left[||q(Z_{(i-1)/N}|Z_{i/N},Z_0) - p_{v_\theta}(Z_{(i-1)/N}|\hat{Z}_{i/N})||_2^2\right], \quad s.t. \quad \mathrm{Law}(Z_{i/N}) \cong \mathrm{Law}(\hat{Z}_{i/N}). \quad (8)$$

For $i \geq 1$, if $\mathrm{Law}(Z_{i/N}) \cong \mathrm{Law}(\hat{Z}_{i/N})$ is guaranteed, then $\mathrm{Law}(Z_{(i-1)/N}) \cong \mathrm{Law}(\hat{Z}_{(i-1)/N})$ can also be guaranteed by optimizing $\mathbb{E}_{\hat{Z}_{i/N},Z_1,Z_0}D_{\mathrm{KL}}(q(Z_{(i-1)/N}|Z_{i/N},Z_0)||p_{v_\theta}(Z_{(i-1)/N}|\hat{Z}_{i/N}))$ in Eq. 8. Based on the prior condition $\mathrm{Law}(Z_1) \cong \mathrm{Law}(\hat{Z}_1)$, we can deduce $\{\mathrm{Law}(Z_{i/N}) \cong \mathrm{Law}(\hat{Z}_{i/N})\}_{i=0}^{N-1}$ sequentially by recursive method.

Note that this derivation via Bayes' Theorem satisfies not only Rectified flow, but also other noise schedules such as VP ODE (Song et al., 2023c; Liu et al., 2022) and VE ODE (Song et al., 2023c; Liu et al., 2022). More details can be found in Appendix M. In fact, the upper bound in Eq. 8 is precisely the optimization objective of FM-KD. The main difference between Eq. 7 and Eq. 8 is that $\hat{Z}_{i/N}$ replaces $Z_{i/N}$, and $\hat{Z}_{i/N}$ is obtained from the reverse sampling process. In this manner, although we increase the computational cost to a certain extent (as sampling is incorporated into the training process), it prevents vanishing gradient in $v_\theta$ and the earlier layers of the student, thereby allowing the distillation process to proceed normally.

## C  LINK FM-KD TO ENSEMBLE

Ensemble is a method that trains multiple models, aggregates their outputs through voting, and produces a final prediction. In this section, we prove theoretically that FM-KD is essentially a unique implicit ensemble method.

First, we define ODE in FM-KD as $X^S - X^T = \frac{dX_t}{dt}$ (for the convenience of derivation, this definition is slightly different from that in the main paper), so we need to fit $||\frac{dX_t}{dt} - g_{v_\theta}(X_t,t)||_2^2$, where $X_t = tX^S + (1-t)X^T, t \sim \mathcal{U}[0,1]$. In inference, this ODE solver defaults to Euler's method in FM-KD, and the sampling must be discrete with $N$ steps because fitting continuous time steps $t$ consumes extensive computational costs. When the meta-encoder $g_{v_\theta}(\cdot)$ is at the optimal solution, we assume that its error from the true value can be expressed as a function of $x_t$ and $t$, and that this function is at 1-Lipschitz.

Thus, we can define $\mathcal{H}(t) = \arg\sup_{X_t}\{||\frac{dX_t}{dt} - g_{v_\theta}(X_t,t)||_2^2\}$, then the truncation error $\mathcal{K}(t)$ can be defined as $\left[\frac{dX_t}{dt} - g_{v_\theta}(\mathcal{H}(t),t)\right]$, which is also at 1-Lipschitz under the assumption $\mathcal{H}(t)$ is at 1-Lipschitz. After that, we also need to define the step number of sampling in inference. We set it as $K$, so $dt$ is $1/K$. Based on the aforementioned notations, we can analyse the truncation error by the recursive method.

For the sake of derivation convenience, we define $\{Z_t\}_t$ as the sampled trajectory in inference to distinguish it from $\{X_t\}_t$ in training. Thus, a step in sampling can be described as $Z_{1-(i+1)/K} =$

$Z_{1-i/K} - g_{v_\theta}(Z_{1-i/K}, 1-i/K)dt$, and $Z_{1-i/K} = X_{1-i/K} + \mathcal{E}(Z_{1-i/K})$, where $\mathcal{E}(Z_{1-i/K})$ refers to the truncation error accumulated to a intermediate sample $Z_{1-i/K}$ in the sampling process. Note that $\mathcal{K}(t)$ and $\mathcal{E}(Z_{1-i/K})$ are not results of the norm, and therefore $\forall t$ and $\forall Z_{1-i/K}$, this derivation does not need to satisfy that $\mathcal{K}(t) \geq 0$ and $\mathcal{E}(Z_{1-i/K}) \geq 0$. This approach avoids the accumulation of the truncation error due to $\ell_2$-norm $\geq 0$. We can derive the sample $Z_{1-(i-1)/K}$ in the next step by the derivation:

$$
\begin{aligned}
Z_{1-(i+1)/K} &= Z_{1-i/K} - (1/K)g_{v_\theta}(Z_{1-i/K}, 1-i/K) \\
&= Z_{1-i/K} - (1/K)g_{v_\theta}(X_{1-i/K} + \mathcal{E}(Z_{1-i/K}), 1-i/K) \\
&= X_{1-i/K} + \mathcal{E}(Z_{1-i/K}) - (1/K)g_{v_\theta}(X_{1-i/K} + \mathcal{E}(Z_{1-i/K}), 1-i/K) \\
&\approx X_{1-i/K} + \mathcal{E}(Z_{1-i/K}) - (1/K)\left[g_{v_\theta}(X_{1-i/K}, 1-i/K) + \mathcal{E}(Z_{1-i/K})\nabla_{X_t}g_{v_\theta}(X_{1-i/K}, 1-i/K)\right], \\
&= X_{1-i/K} + \mathcal{E}(Z_{1-i/K}) - (1/K)\left[g_{v_\theta}(X_{1-i/K}, 1-i/K) + \mathcal{E}(Z_{1-i/K})\psi(1-i/K)\right],
\end{aligned}
$$
$$(9)$$

where $\psi(t) = \nabla_{X_t}g_{v_\theta}(X_t, t)$. Then, Eq. 9 can continue to be derived as

$$
\begin{aligned}
Z_{1-(i+1)/K} &\approx X_{1-(i+1)/K} + \mathcal{E}(Z_{1-i/K}) + (1/K)\mathcal{K}(1-i/K) - (1/K)\mathcal{E}(Z_{1-i/K})\psi(1-i/K) \\
&= X_{1-(i+1)/K} + \mathcal{E}(Z_{1-i/K})[1-(1/K)\psi(1-i/K)] + (1/K)\mathcal{K}(1-i/K) \\
Z_{1-(i+1)/K} - X_{1-(i+1)/K} &= \mathcal{E}(Z_{1-i/K})[1-(1/K)\psi(1-i/K)] + (1/K)\mathcal{K}(1-i/K).
\end{aligned}
$$
$$(10)$$

Thus, $\mathcal{E}(Z_{1-(i+1)/K}) = \mathcal{E}(Z_{1-i/K})[1-(1/K)\psi(1-i/K)] + (1/K)\mathcal{K}(1-i/K)$. After that, the recursive method leads us to the following conclusions:

$$\mathcal{E}(Z_{1-1/K}) = (1/K)\mathcal{K}(1)$$
$$\mathcal{E}(Z_{1-2/K}) = \mathcal{E}(Z_{1-1/K})(1-(1/K)\psi(1-1/K)) + (1/K)\mathcal{K}(1-1/K)$$
$$\vdots$$

$$
\mathcal{E}(Z_0) = (1/K)\left[\sum_{i=0}^{K-1}\mathcal{K}(1-i/K)\right] + (1/K^2)\left[\sum_{j=1}^{K-1}\left[\psi(1-j/K)\left(\sum_{i=0}^{j-1}\mathcal{K}(1-i/K)\right)\right]\right] + \mathcal{O}(1/K^3).
$$
$$(11)$$

Looking at the first term, we can see that the truncation error comes from summing $\mathcal{K}(\cdot)$ over all time points. When treating the error sampling as Monte Carlo sampling, with a sufficient number of samplings $K$, it becomes possible for FM-KD to approximate ensemble methods and thus estimate the ground truth effectively.

## D PAIR DECOUPLING

In this section, we present Pair Decoupling (PD), a straightforward yet effective technique for enhancing performance in the feature-based distillation scenario of image classification using FM-KD. This method involves shuffling a subset of samples in a batch to achieve regularization, thereby preventing overfitting of the teacher's refined low-level hierarchical features. Let $B$, $C$, $H$ and $W$ denote the batch size, the number of channels, the height of the feature map, and the width of the feature map, respectively. Given the teacher's feature map $X^T \in \mathbb{R}^{B \times C \times H \times W}$ from a specific layer, PD is applied prior to all FM-KD related calculations. Implementing PD involves defining a hyperparameter, the dirac ratio $\beta_d$, and perturbing $B - \lfloor\beta_d B\rfloor$ samples in the batch. The Pytorch code for this is provided in Appendix A. Specifically, PD selects the random $B - \lfloor\beta_d B\rfloor$ samples $X^T[0 : B - \lfloor\beta_d B\rfloor]$ in a batch and then shuffles them:

$$X^T[0 : B - \lfloor\beta_d B\rfloor] = \mathbf{shuffle}\left(X^T[0 : B - \lfloor\beta_d B\rfloor]\right).$$

We refer to the hyperparameter $\beta_d$ as "dirac ratio" because, following the PD operation, $\lfloor\beta_d B\rfloor$ samples are used to compute $\mathcal{L}_{\text{FM-KD}}$. Here, $X^T$ and $X^S$ are treated as Dirac distributions with the objective of achieving one-to-one matching. Conversely, the remaining $B - \lfloor\beta_d B\rfloor$ samples are utilized in the computation of $\mathcal{L}_{\text{FM-KD}}$, where $X^T$ and $X^S$ are considered as non-Dirac distributions, targeting many-to-many matching.

Due to the specificity of the feature-based distillation scenario for image classification, PD is specifically designed to avoid over-matching the refined low-level feature thus improving the final performance of the student. Meanwhile, our experiments in Appendix E empirically demonstrate that PD

is effective only in the feature-based distillation scenario of image classification, whereas in other scenarios it rather degrades the performance. This is because matching the teacher's feature/logit at a fine-grained level is closely related to the final performance of the student in the logit-based distillation scenario for image classification as well as the feature-based distillation scenario for object detection. In other words, in the feature-based distillation scenario for image classification, it does not imply that improving similarity between the student's low-level feature and the teacher's low-level feature will result in the greater classification accuracy of the student.

## E ADDITIONAL ABLATION EXPERIMENT

Here, we experimentally substantiate some empirical findings on the topics of normalization layer selection in the meta-encoder, where stages used for distillation in the feature-based scenario, and ideal configuration of dirac ratio $\beta_d$ in different scenarios.

Ablation experiments on various normalization operations reveal instability in the FM-KD training paradigm when using BatchNorm. As observed in Table 6, the accuracy achieved with BatchNorm as the normalization layer is approximately 1%, even when the training of the student remains stable (*i.e.*, the loss is not NAN). This indicates that using BatchNorm in FM-KD introduces instability by computing the mean and variance of inputs at different time points during inference. It is essential to note that although BatchNorm is applied in DiffKD, this choice is justified as the student converges effectively with a sufficiently high number of layers in the Diffusion Model (*i.e.* meta-encoder) mentioned in their work. Similar results are obtained in our studies by replacing the meta-encoder in FM-KD with Diffusion Model in DiffKD.

| Normalization type | GroupNorm | BatchNorm |
|---|---|---|
| WRN-40-2 (T) | 75.61 | 75.61 |
| WRN-40-2 (S+Baseline) | - | 73.26 |
| WRN-40-2 (S+DIST) | - | 75.29 |
| WRN-16-2 (S+FM-KD $K$=1) | 75.58 | 1.00 |
| WRN-16-2 (S+FM-KD $K$=2) | 75.85 | 1.00 |
| WRN-16-2 (S+FM-KD $K$=4) | 75.87 | 1.10 |
| WRN-16-2 (S+FM-KD $K$=8) | 75.87 | 1.43 |

Table 6: Experiments were conducted on the different normalization type of FM-KD on CIFAR-100. Note that in this table all the architecture of the meta-encoder and the form of the loss function are set as CNN and DIST, respectively.

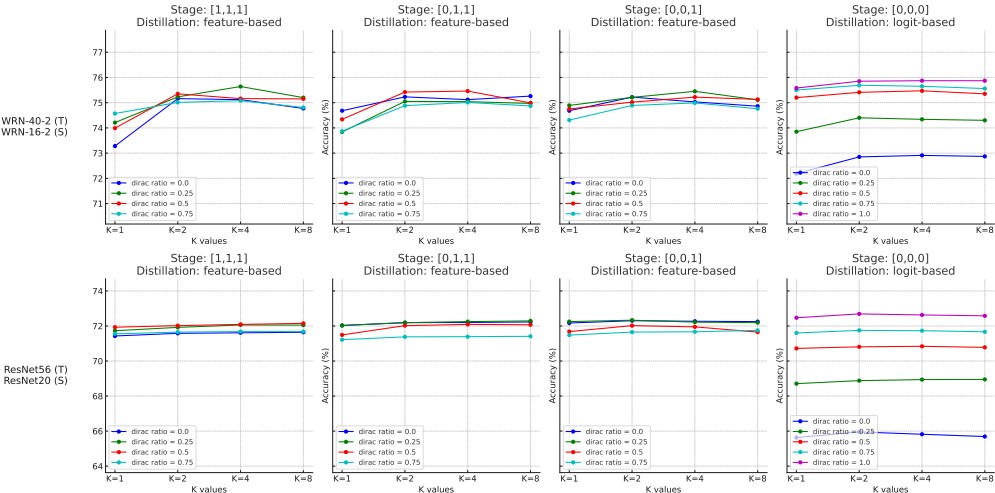

Figure 7: Experiments were conducted on the various hyperparameter $\beta_d$ (dirac ratio) and different distillation stages of FM-KD on CIFAR-100. Note that in this figure all the architecture of the meta-encoder and the form of the loss function are set as CNN and DIST, respectively. The choice of the last three stages was fixed for distillation analysis. In the figure, the notation "[n3, n2, n1]" indicates that n$x$=1 signifies the use of that the $x$th last stage for distillation. For example, "[0, 1, 1]" signifies the utilization of the 1st and 2nd last stages for distillation.

In Fig. 7, we investigated the optimal stages for distillation in the feature-based scenario and the ideal configuration for the dirac ratio $\beta_d$. As there are no specific distillation stages for the logit-based distillation, we designated it as "[0, 0, 0]" for clarity. Our observations indicate that in the feature-based distillation scenario, the distillation stage does not significantly affect the final outcomes. Meanwhile, the configuration "[1, 1, 1]" often underperforms compared with "[0, 1, 1]" and "[0, 0, 1]". This observation aligns with the conclusions drawn from most of the prior feature-based

distillation studies (Tung & Mori, 2019; Zagoruyko & Komodakis, 2016a; Zong et al., 2023). Moreover, for different values of $\beta_d$, the settings $\beta_d$=0.25 typically yields the best result in feature-based distillation, while $\beta_d$=1.0 excels in the logit-based distillation. This implies that the PD technique is more effective in the feature-based distillation context for image classification, and less so in the logit-based distillation.

It is worth noting that our experiments on PD in object detection revealed that $\beta_d$=1.0 and $\beta_d$=0.75 yield comparable performance, whereas a decrease in $\beta_d$ results in diminished performance. Based on these findings, we recommend using $\beta_d$=0.25 as the default in the feature-based distillation scenario for image classification and $\beta_d$=1.0 in other contexts.

## F   ADDITIONAL RELATED WORK ON CONTINUOUS NETWORK REPRESENTATIONS

With the development of deep learning, there are a number of architectures belonging to continuous network representation, such as RNN (Williams & Zipser, 1989), LSTM (S & J, 1997), Neural ODE (Chen et al., 2018), GflowNet family (Bengio et al., 2021; Zhang et al., 2022), diffusion model family (Song et al., 2023c; Karras et al., 2022; Ho et al., 2020; Song et al., 2023a), INN (Solodskikh et al., 2023) and DiffKD (Huang et al., 2023). Here, we mainly emphasize the similarities and differences between our proposed FM-KD and these methods. In this way, we show the novelty of the FM-KD design and its advantages in application:

- The application scenario of FM-KD is different from RNN, LSTM, Neural ODE, GflowNet family, diffusion model family and INN. Of these, only our method and DiffKD are applied to knowledge distillation in the form of continuous network representations.

- RNN, LSTM, Neural ODE, GflowNet family, diffusion model family, INN, and FM-KD have a meta-encoder shared parameters. However, the difference is that the forward process (meaning the backward process in diffusion model family and FM-KD) in RNN, LSTM, and GflowNet family is unknown. Unlike Neural ODE, diffusion model family and FM-KD, there exists a human-designed sampling process (*a.k.a* predefined forward processes), which makes it impossible to use numerical integration to trade-off performance and efficiency.

- INN primarily enables the continuous representation of convolutional operations (*i.e.* convolutional kernels), not the entire network. In contrast, the Neural ODE, diffusion model family, continuously represents the entire network.

- The primary distinction between Neural ODE/FM-KD and the diffusion model family lies in their training paradigms. The diffusion model family is trained using unpaired samples, aiming to capture the entire data distribution. In contrast, Neural ODE/FM-KD utilizes paired samples, focusing on learning the Dirac distribution of the output.

- The biggest difference between FM-KD and Neural ODE is that FM-KD has a deterministic a priori forward process to model the optimization objective of the intermediate points (*i.e.* $\{Z_{1-i/N}\}_i$), which ensures the stability of training. Neural ODE has no such a priori forward process, and simply expects the network itself to learn a continuous representation from input to output.

## G   STRONGER STRATEGIES AND STRONGER TEACHER COMPARISON

In recent years, with the advancement of deep learning, stronger training strategies and higher-quality foundational models have emerged. As a result, traditional distillation methods are no longer sufficient for capturing a superior student. In this context, we utilize the ResNet50 (with an accuracy of 80.1%) from TIMM (Wightman et al., 2021) training as a stronger teacher to distill the ResNet18. Simultaneously, we adopt some stronger strategies: the learning rate begins at 5e-4, the chosen optimizer is AdamW, the batch size is set as 1024, the number of training epochs is set as 350, the learning rate warms up over 3 epochs, and then decays at a rate of 0.9874 per epoch. For data augmentation, we employ a combination of RandomCrop, RandomClip, RandAugment (Cubuk et al., 2020), and RandomErasing (Zhong et al., 2020). It's important to note that the loss function

and the meta-encoder in FM-KD remain consistent with the main paper, being DIST and Swin-Transformer, respectively. Finally, the experimental results on ImageNet-1k are presented in Table 7.

| | Teacher (ResNet50) | *DIST* | FM-KD ($K$=1) | FM-KD ($K$=2) | FM-KD ($K$=4) | FM-KD ($K$=8) |
|---|---|---|---|---|---|---|
| Top-1 Acc. | 80.12% | *72.89%* | 72.61% | 73.11% | 73.59% | 73.71% |

Table 7: Additional results of FM-KD in the stronger strategies and stronger teacher setting.

From Table 7, we can observe that FM-KD performs remarkably when both the teacher and the strategies are stronger. For instance, when $K$=8, the student's accuracy is 73.70%, which is 0.82% higher than the baseline *DIST*. This is a clear indication that FM-KD can be generalized to scenarios with strong strategies and a stronger teacher.

## H    VISUALIZATION OF SAMPLING TRAJECTORY

To elucidate the sampling mechanism of FM-KD, we utilize the student obtained by training the ResNet34-ResNet18 pair on ImageNet-1k and visualize the student output's sampling trajectory (*i.e.*, $\{Z_{1-i/K}\}_{i=1}^{K}$, where $K$ is set as 8) during inference. Note that the loss function and meta-encoder are set as DIST and Swin-Transformer in training, respectively. Since general visualization methods are designed for feature maps in intermediate layers, it's challenging to demonstrate that better visualization directly correlates with improved image classification performance. Therefore, we employ the reliability histogram for the visualization of the sampling trajectory, thereby demonstrating that FM-KD, as an implicit ensemble method, indeed enhances the generalization ability of the student.

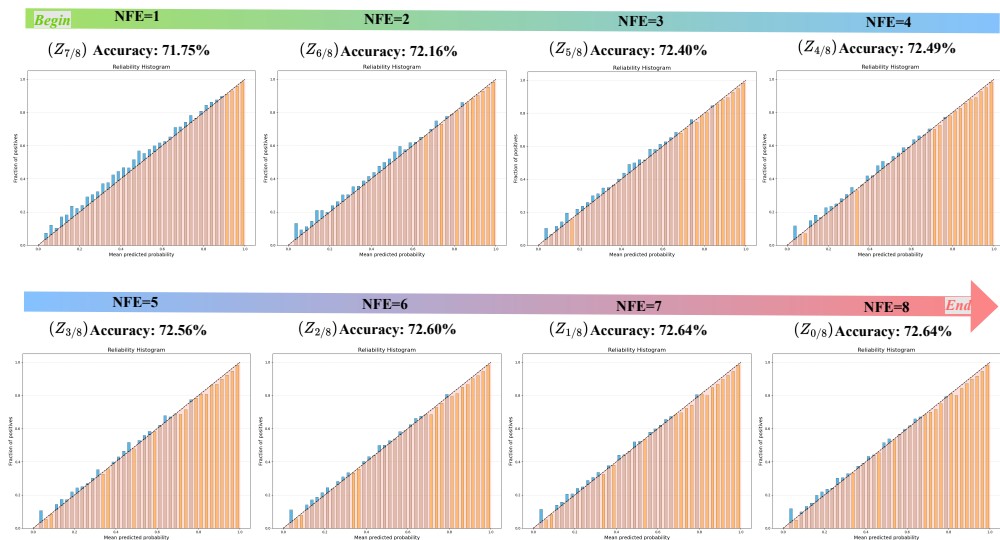

Figure 8: The visualization of the sampling trajectory in the trained ResNet18.

The reliability histogram typically plots predicted probabilities on the x-axis and the fraction of positives on the y-axis. Typically, the closer the predicted probability is to the fraction of positives, the better the student's prediction. Therefore, the closer the peak of the student's reliability histogram bin is to the diagonal, the stronger its generalization ability. Thus, it is clear from Fig. 8 that the reliability histogram is not well-presented at the beginning (*i.e.* $Z_{7/8}$) of the sampling trajectory. As $i$ in $Z_{1-i/K}$ gradually decreases, the representation of its reliability histogram improves, which indicates that both the generalization ability and the reliability of the student are enhanced.

## I    IMPLEMENTATION DETAIL

### I.1    TRAINING STRATEGIES

We train on image classification datasets including CIFAR-100 (Krizhevsky et al., 2009) and ImageNet-1k (Russakovsky et al., 2015), and object detection dataset including MS-COCO (Lin et al., 2014). **For Offline KD**, the training strategy of image classification follows CRD (Tian et al., 2019) and DIST, while the training strategy of object detection follows PKD. Specifically, for CIFAR-100, the learning rate is 0.05 (when MobileNetV2 or ShuffleNetV1 as the student, the learning rate is 0.01), batch size is 64, total number of epochs is 240, and the learning rate is linearly reduced to 0.1 of its previous value at epochs 150, 180, and 210; for ImageNet-1k, the training learning rate is 0.1, batch size is 256, total number of epochs is 100, and the learning rate is linearly reduced to 0.1 of its previous value at epochs 30, 60, and 90; for MS-COCO, the training learning rate is 0.02, batch size is 16, total number of epochs is 24, and the learning rate is linearly reduced to 0.1 of its previous value at epochs 16 and 22. **For Online KD**, all hyperparameters settings follow AHBF-OKD (Gong et al., 2023) and are unchanged. For conviction, we report the mean test accuracy with 3 runs for all experimental results.

## I.2 LOSS FUNCTION AND META-ENCODER

The loss weights of FM-KD and its variant OFM-KD are not explicitly set, and their values will follow the loss weight settings of the metric-based distillation method introduced by themselves. For instance, if FM-KD applies DIST as its $L(\cdot, \cdot)$, the loss weights $\beta$ and $\gamma$ are both set to 2 as mentioned in the original paper. For convenience of description, all forms "FM-KD ($K$=number)" or "OFM-KD ($K$=number)" refer to the corresponding algorithms that sampled "number" steps during inference.

For all comparative experiments on CIFAR-100, FM-KD and OFM-KD use Swin-Transformer as the meta-encoder and DIST as the metric-based distillation method, except for VGG13-VGG8 and VGG13-MobileNetV2 pairs in the Offline KD scenario. VGG13-VGG8 and VGG13-MobileNetV2 pairs in the Offline KD scenario use Swin-Transformer as the meta-encoder and DKD as the metric-based distillation method. For all comparative experiments on ImageNet-1k, in the Offline KD scenario, FM-KD uses MLP (*i.e.* 2-MLP) as the meta-encoder and DIST as the metric-based distillation method; in the Online KD scenario, OFM-KD uses Swin-Transformer as the meta-encoder and DIST as the metric-based distillation method.

For FM-KD$^\Theta$, the loss function and meta-encoder are set to DKD and Swin-Transformer with all pairs on CIFAR-100; the loss function and meta-encoder are set to DIST and MLP with all pairs on ImageNet-1k; the balance weight $\alpha^\Theta$ is set as 1.0, 1.0 and 0.0 on all teacher-student pairs on CIFAR-100, ResNet34-ResNet18 pair on ImageNet-1k and ResNet50-MobileNetV2 pair on ImageNet-1k, respectively.

For object detection, unless otherwise specified, FM-KD uses CNN as the meta-encoder and PKD as the metric-based distillation method.

For the architecture of the meta-encoder, we adopt a task-specific setup. Swin-Transformer adopts one layer of **[Swin Attention-Linear-ReLU-Linear]** in the Offline KD scenario, and the number of heads is 4. In the Online KD scenario, if the student architecture is not ResNet18 then we add the same extra layer in the meta-encoder. CNN uses one layer of **[SiLU-Conv-GroupNorm-SiLU-Conv]** in the image classification datasets and two layer of **[Depthwise Conv-LayerNorm-Pointwise Conv-GeLU-Pointwise Conv]** in the object detection dataset. In image classification, the kernel size of first convolutional layer is 3×3, and the second layer is 1×1. And in object detection, the kernel size of depthwise convolutional layer is 7×7. MLP adopts two layers of **[Linear-ReLU-Linear]** in the logit-based distillation scenario and one layer of **[Linear-ReLU-Linear]** in the feature-based distillation scenario. Besides, the shape transformation function $\mathcal{T}(\cdot)$ utilizes one layer of **[Conv]** or **[Identity Function]** (if no shape alignment is required) in the feature-based distillation scenario, and we use one layer of **[AdaptAvgpool(1)-Linear]** in the logit-based distillation scenario. Note that in the logit-based distillation scenario, FM-KD completes flow matching on the logit, so **[AdaptAvgpool(1)-Linear]** essentially represents the classification layer.

## J ADDITIONAL TRAINING AND INFERENCE COMPUTATIONAL COST DISCUSSION

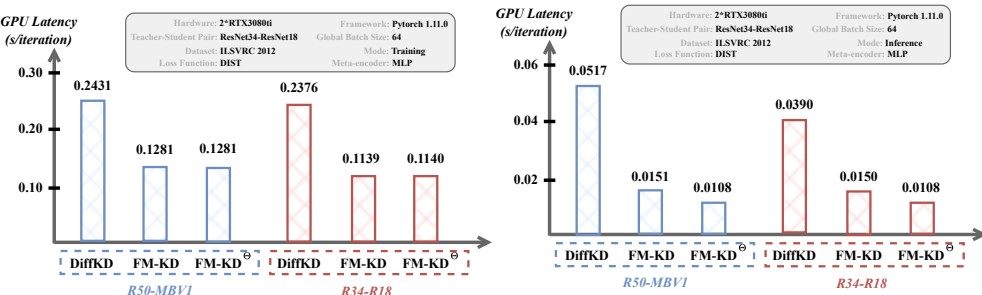

Figure 9: Training and Inference computational cost of DiffKD, FM-KD and FM-KD$^\Theta$

Our proposed FM-KD, similar to DiffKD, incurs an additional computational burden during inference. However, our variant, FM-KD$^\Theta$, differs in that it avoids this extra computational load during inference. This is achieved by transferring the knowledge in $Z_0$ (*w.r.t.* $t=0$) in FM-KD to the vanilla classification head of the student. To provide a clear comparison of the computational costs of DiffKD, FM-KD, and FM-KD$^\Theta$, we have conducted relevant measurements. The results are presented in Fig. 9. Notably, both FM-KD and FM-KD$^\Theta$ utilize logit-based distillation as the loss function (referred to as DIST) and employ a 2-layer MLP as the meta-encoder. Moreover, DiffKD adheres to the approach outlined in its original paper, employing both feature-based and logit-based distillation. The feature-based distillation in DiffKD, which relies on Bottleneck from ResNet, is implemented in the meta-encoder and is inserted into the backbone output feature before average pooling. Meanwhile, its logit-based distillation employs a 1-layer MLP as the meta-encoder and is inserted into the output logit of the classification head. As presented in Fig. 9, the computational overhead of DiffKD, in both training and inference, is drastically higher than that of FM-KD and FM-KD$^\Theta$. Furthermore, FM-KD$^\Theta$ aligns with classical knowledge distillation algorithms in terms of computational cost during inference, offering additional savings in inference overhead compared to FM-KD.

## K  BEST META-ENCODER CHOICE ON IMAGENET-1K

As illustrated in Figure 5, FM-KD achieves the highest effectiveness and efficiency on ImageNet-1k when implemented with MLP. Accordingly, this section presents the optimal performance of FM-KD by using MLP for meta-encoder on ImageNet-1k and examines the impact of varying the number of MLP layers on its performance.

| Method | FM-KD | FM-KD | FM-KD | DiffKD |
|---|---|---|---|---|
| Meta-encoder | 1-MLP | 2-MLP | 3-MLP | 2-BottleNeck+ Conv+BN+MLP |
| ResNet34-ResNet18 | 72.48 | 73.17 | **73.28** | 72.49 |
| ResNet50-MobileNetV1 | 73.74 | 74.22 | **74.28** | 73.78 |

Table 8: The influence of the number of layers in the meta-encoder (*i.e.* MLP) on student performance on ImageNet-1k. Note that all results from FM-KD are obtained when $K$=8.

The experimental results presented in Table 8 show that FM-KD outperforms DiffKD with 2-later MLP (*i.e.* 2-MLP). Furthermore, as detailed in Appendix J, the training and inference costs of FM-KD are nearly half those of DiffKD. This effectively demonstrates FM-KD's capability to not only outperform DiffKD but also achieve state-of-the-art performance. Meanwhile, the performance of FM-KD improves as the number of layers in the MLP increases.

## L  ARCHITECTURE-SENSITIVE EXPERIMENTS BETWEEN FM-KD AND DIFFKD

In order to know the sensitivity of FM-KD and DiffKD to architecture and for further fair comparisons, we perform DiffKD to use 2-MLP from FM-KD as its meta-encoder, and FM-KD to use 2-Bottleneck+Conv+BN+MLP from DiffKD as its meta-encoder. The experiments were then conducted on ImageNet-1k using ResNet50-MobileNetV1 pair. Unfortunately, when employing the

| DiffKD uses FM-KD's meta-encoder (*i.e.* 2-MLP) | FM-KD uses DiffKD's meta-encoder (*i.e.* 2-Bottleneck+Conv+BN+MLP) | DiffKD uses DiffKD's meta-encoder (*i.e.* 2-Bottleneck+Conv+BN+MLP) | FM-KD uses FM-KD's meta-encoder (*i.e.* 2-MLP) |
|---|---|---|---|
| NAN | **74.26%** | 73.78% | 74.22% |

Table 9: Comparison experimental result between FM-KD and DiffKD with ResNet50-MobileNetV1 pair on ImageNet-1k. Note that the results from FM-KD are obtained when $K$=8.

logit-based distillation approach of DiffKD (following its official code and implementation (Huang et al., 2023)), its loss became NAN at epoch 1. However, in Table 9, we discover that the result (with $K$=8) of FM-KD using 2-Bottleneck+Conv+BN+MLP from DiffKD as its meta-encoder, which significantly outperformed DiffKD.

## M  UNIFY VP SDE, VE SDE AND RECTIFIED FLOW IN FM-KD

Vanilla diffusion processes such as VP SDE (Song et al., 2023c) and VE SDE (Song et al., 2023c) can be transformed into the flow form proposed in our work. The reason for adopting Rectified flow in our main paper is due to its simplicity in implementation and understanding. Moreover, 1) in the derivation of approximating ensembles (Proposition 3.2), it can be proved that the truncation error at each time step has the same impact on the ultimate error (*i.e.*, equal weight), and 2) Rectified flow enhances the student performance by its accelerated sampling property when NFE is very small. Specifically, Rectified flow has the ability to minimize the hessian matrix (Lee et al., 2023) with respect to $Z_t$, which enables the estimation $g_{v_\theta}(Z_t, t)$ of $dZ_t$ to also accurately estimate $\{dZ_{t-\Delta_t}, dZ_{t-2\Delta_t}, \cdots, dZ_s\}$, where $t$ and $s$ refer to the source and target time points, respectively, ultimately reducing the truncation error of $Z_t + \int_t^s g_{v_\theta}(Z_\tau, \tau)d\tau$. This point is demonstrated by the experiments in papers (Liu et al., 2022; Lee et al., 2023).

To better understand this, we present the unified modeling form of FM-KD, which can simultaneously hold VP SDE, VE SDE and Rectified flow. Note that both VP SDE and VP SDE can be transformed into ODE form, referring to deterministic forward and backward processes. Here, the ODE forms of VP SDE and VE SDE are named VP ODE and VE ODE.

All probability flows can be written in the following form:

$$Z_t = \alpha_t X_S + \sigma_t X_T, \ s.t. \ Z_0 \approx X_T, Z_1 \approx \alpha_1 X_S, \lim_{t \to 0} \alpha_t = 0, \lim_{t \to 0} \sigma_t = 1. \tag{12}$$

The training paradigm of them can be denoted as

$$\arg\min_{v_\theta} \mathbb{E}_{(Z_1, Z_0, t)} \|g_{v_\theta}(Z_t, t) - \nabla_t Z_t\|_2^2$$
$$= \arg\min_{v_\theta} \mathbb{E}_{(Z_1, Z_0, t)} \|g_{v_\theta}(Z_t, t) - (\nabla_t \alpha_t Z_1 + \nabla_t \sigma_t Z_0)\|_2^2. \tag{13}$$

**VP ODE:**  (1) $\alpha_t = \exp(-\frac{1}{4}a(1-t)^2 - \frac{1}{2}b(1-t))$; (2) $\sigma_t = \sqrt{1 - \alpha_t^2}$, $s..t.$  $a = 19.9, b = 0.1$.

**VE ODE:**  (1) $\alpha_t = a(\frac{b}{a})^t$; (2) $\sigma_t = 1$, $s.t.$  $a = 0.02, b = 100$.

**Rectified flow:**  (1) $\alpha_t = t$; (2) $\sigma_t = 1 - t$.

Substituting $\alpha_t$ and $\sigma_t$ yields:

**VP ODE:**

$$\arg\min_{v_\theta} \mathbb{E}_{(Z_1, Z_0, t)} \|g_{v_\theta}(Z_t, t) - ((\frac{1}{2}a(1-t) + \frac{1}{2}b)\alpha_t Z_1 - \frac{\alpha_t}{\sqrt{1 - \alpha_t^2}}\alpha_t(\frac{1}{2}a(1-t) + \frac{1}{2}b)Z_0)\|_2^2. \tag{14}$$

**VE ODE:**

$$\arg\min_{v_\theta} \mathbb{E}_{(Z_1, Z_0, t)} \|g_{v_\theta}(Z_t, t) - (\alpha_t[\log(b) - \log(a)]Z_1)\|_2^2. \tag{15}$$

**Rectified flow:**

$$\arg\min_{v_\theta} \mathbb{E}_{(Z_1, Z_0, t)} ||g_{v_\theta}(Z_t, t) - (Z_1 - Z_0)||_2^2. \tag{16}$$

All forms can be transformed into serial training forms by Theorem 3.1 in our paper[2]:

$$\mathcal{L}_{\text{FM-KD++}} = \mathbb{E}_{(X^S, X^T, Y)} \frac{1}{N} \sum_{i=0}^{N-1} L(\mathcal{T}((\nabla_t \alpha_t Z_1 - g_{v_\theta}(Z_{1-i/N}, 1-i/N))/ - \nabla_t \sigma_t), X^T)$$

$$+ \underbrace{L(\mathcal{T}((\nabla_t \alpha_t Z_1 - g_{v_\theta}(Z_{1-i/N}, 1-i/N))/ - \nabla_t \sigma_t), Y)}_{\text{match the ground truth label (optional)}}, \tag{17}$$

the sampling process: $Z_{1-i/N} = Z_{1-(i-1)/N} - g_{v_\theta}(Z_{1-(i-1)/N}, 1-(i-1)/N)/N, \quad s.t. \quad i \geq 1,$

where $Z_1 = \alpha_1 X_S$. Thus, the key to achieving knowledge transfer in knowledge distillation is not Rectified flow, but the form of deterministic sampling in both the forward and backward processes and the serial training paradigm given in Theorem 3.1 of our paper.

**Evaluation.** In the practical implementation, since $\lim_{t \to 1} \nabla_t \alpha_t = +\infty$ in VP ODE, and considering that $\nabla_t \alpha_t$ and $\nabla_t \sigma_t$ show large variations at different $t$ in both VP ODE and VE ODE, both VP ODE and VE ODE are expressed in the forms of differentiations $\frac{\alpha_t - \alpha_{t-\Delta t}}{t - \Delta t}$ and $\frac{\sigma_t - \sigma_{t-\Delta t}}{t - \Delta t}$. Since $\nabla_t \sigma_t \equiv 0$ in VE ODE cannot be divided, we modified $\sigma_t$ from $\sigma(t) = 1$ to $\sigma(t) = 1 - 0.1t$. In addition, our experiments revealed instability in the flow loss of VE ODE and VP ODE training, necessitating the use of the learning rate warm-up technique (extending up to 20 epochs) for effective training. And $b$ in VE ODE is extra reduced to 10. The test accuracy per epoch for VP ODE, VE ODE and Rectified flow (*i.e.*, the default form used in our paper) is illustrated in Fig. 10.

Different Noise Schedule Experimental Result Visualization

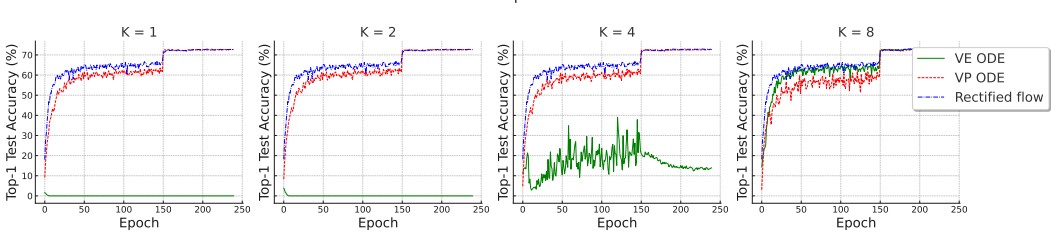

Figure 10: Trajectories of top-1 test accuracy with WRN-40-2-WRN-16-2 pair on CIFAR-100 for various noise schedules: VP ODE, VE ODE, and Rectified flow.

The experimental results in Fig. 10 are obtained on CIFAR-100 with WRN-40-2-WRN-16-2 pair. VP ODE, VE ODE, and Rectified flow all utilize 2-MLP as the meta-encoder, and DIST as the loss function (modifying the hyperparameter temperature to 1 for stable training). It can be observed that the training paradigm proposed in Eq. 17 is capable of effectively training all noise schedules. In particular, Rectified flow is comparatively more stable and efficient than VP ODE and VE ODE.

---

[2]For convenience, we ignore time steps here. It is worth noting that, due to the adaptability of step size in the Euler method, introducing this hyperparameter is entirely feasible.

