# OpenReview forum: "Knowledge Distillation via Flow Matching"
_ICLR.cc/2024/Conference — Submitted to ICLR 2024_

### Official Review · Reviewer_c4jx · 2023-10-30

**Soundness:** 3 good
**Presentation:** 2 fair
**Contribution:** 2 fair
**Rating:** 5
**Confidence:** 5

**Summary:**

In this paper,  the authors introduce rectified flow into knowledge distillation and leverage multi-step sampling strategies to achieve precision flow matching. It can be integrated with metric-based distillation methods. The authors offered theoretical analysis which demonstrates that the training objective of FM-KD is equivalent to minimizing the upper bound of the teacher feature map's or logit's negative log-likelihood.  And the authors also offered an online distillation version.

**Strengths:**

1. Distillation is an important topic to our community, the proposed method is simple.
2. Some of the "same architecture setting" results are promising.
3. The write-up is easy to understand.

**Weaknesses:**

My major concern is the generalization, given that distillation is a well-defined topic, but this method seems like doesn't work well in heterogeneous architecture settings. Especially when there is a big difference in size between teacher models and student models.

**Questions:**

Can you explain or verify why this method can't work well when the gap is big between teacher and student?

---

> ### Author Response · Authors · 2023-11-19
> **Response to Reviewer c4jx**
>
> Thank you for your valuable comments, we exert our greatest effort to address your comments.
>
> ### **Q1. My major concern is the generalization, given that distillation is a well-defined topic, but this method seems like doesn't work well in heterogeneous architecture settings. Especially when there is a big difference in size between teacher models and student models.**
>
> **A1.** Thank you for your valuable suggestions.
>
> In our experiments, the heterogeneous teacher-student pairs, ResNet50-MobileNetV1 on ImageNet-1k and VGG13-MobileNetV2 on CIFAR-100, did not achieve SOTA accuracy. This outcome can be attributed to suboptimal configurations of the loss function and meta-encoder for FM-KD, rather than suggesting that FM-KD is ineffective in heterogeneous architecture settings.
>
> We experiment with the optimal settings, and the additional experimental results are presented in Table I and II. We can observe that FM-KD beats all previous methods and achieves SOTA performance.
>
> Table I (ImageNet-1k, loss function is set as DIST, K=8)
>
> | T-S pair | Meta-encoder | R50-MBV1 |
> | ---- | ---- | ---- |
> | Top-1 Acc. (K=8) | 2-MLP | 74.22 |
> | Top-1 Acc. (K=8) | 3-MLP | 74.28 |
>
> Table II (CIFAR-100, meta-encoder is set as Swin-Transformer, loss function is set as DKD)
>
> | T-S pair | VGG13-VGG8 | VGG13-MBV2 |
> | ---- | ---- | ---- |
> | Top-1 Acc. (K=1) | 75.21 | 69.68 |
> | Top-1 Acc. (K=2) | 74.86 | 69.52 |
> | Top-1 Acc. (K=4) | 75.42 | 69.94 |
> | Top-1 Acc. (K=8) | 74.46 | 69.94 |
>
> ### **Q2. Can you explain or verify why this method can't work well when the gap is big between teacher and student?**
>
> **A2.** Thank you for your interesting question. FM-KD is effective in your mentioned scenarios. Specifically, as shown in Tables I and II, FM-KD is effective in heterogeneous architecture settings with a big gap between the teacher and the student. In scenarios where a bigger gap exists between the teacher and the student, we conduct experiments using TIMM's ResNet50 as the teacher and ResNet18 as the student. The experimental results in Table III (refer to Appendix G) indicate that FM-KD remains effective.
>
> Table III (ImageNet-1k, epoch is set as 350, teacher is set as ResNet 50 from TIMM, student is set as ResNet18)
>
> | Model |  ResNet50 | ResNet18 (DIST) | ResNet18 (FM-KD, K=8) |
> | ---- | ---- | ---- | ---- |
> | Type | Teacher | Student | Student |
> | Top-1 Acc. | 80.12 | 72.89 | 73.71 |

---

### Official Review · Reviewer_6oRH · 2023-10-31

**Soundness:** 3 good
**Presentation:** 3 good
**Contribution:** 3 good
**Rating:** 5
**Confidence:** 4

**Summary:**

This paper proposes a new knowledge distillation method FM-KD that combines rectified flow with knowledge distillation. It can be combined with any metric approach and meta-encoder structure, and can be converted to online distillation with minor modifications. The FM-KD method achieves interesting experimental results on the CIFAR-100, ImageNet-1k, and MS-COCO datasets.

**Strengths:**

•	The FM-KD method is flexible enough to be combined with any metric approach and meta-encoder structure, and can also be converted to an online distillation framework.
•	There are theoretical analyses to support the rationality of the methodological design.

**Weaknesses:**

•	This paper is similar to DiffKD where it is a combination of generative model and knowledge distillation, except that it replaces the diffusion model with the rectified flow, which has limited innovation.
•	The FM-KD method proposed in this paper modifies the structure of the student network and increases the cost during model inference, which is a shortcoming of a model compression algorithm. The computational cost during inference is not given in the paper, and if refer to Fig. 5, the increase in cost is obvious and there is some unfairness in comparison with other methods. Are there still significant advantages of the FM-KD approach over baseline models that increase the number of parameters to make time-consuming approximations?
•	The experimental performance is not outstanding enough. There is no significant advantage over other knowledge distillation methods in Table 3 of the experiments for object detection, and there are also several tests in the image classification experiments that do not perform as well as other existing methods.

**Questions:**

see the weakness section.

---

> ### Author Response · Authors · 2023-11-19
> **Response to Reviewer 6oRH (Part I)**
>
> Thank you for reviewing our paper. Below, we provide responses to your comments and queries.
>
> ### **Q1. This paper is similar to DiffKD where it is a combination of generative model and knowledge distillation, except that it replaces the diffusion model with the rectified flow, which has limited innovation.**
>
> **A1.** Thank you for your valuable feedback. Our approach does not merely substitute the diffusion model with Rectified Flow or directly apply Rectified Flow for knowledge transfer. Specifically, DiffKD employs the original training loss of the diffusion model (i.e., $L_{diff}$) and combines it with two additional auxiliary losses (i.e., $L_{ae}$ and $L_{diffkd}$) as detailed in its original paper for training. By contrast, we utilize our proposed training loss $L_{FM-KD}$ (solely one), guaranteed by Theorem 3.1 in the original/revised paper, which differs from each of $L_{diff}$, $L_{ae}$, and $L_{diffkd}$. Moreover, from Eq. 2 (i.e. original Rectified flow loss function) to Eq. 3 (i.e. our proposed serial loss function) in our original/revised paper we implement some critical improvements. Without these improvements, Rectified flow (w.r.t. Eq. 2) in the WRN-40-2-WRN-16-2 pair on CIFAR-100 would have encountered the problem of vanishing gradients, resulting in a final test accuracy 3.42%.
>
> It might be helpful to provide a clear outline of our contributions to ensure a better understanding.
>
> 1. We introduce Rectified flow instead of diffusion processes since Rectified flow is able to establish a bridge between two empirical distributions (teacher feature/logit distribution and student feature/logit distribution) and is well suited for knowledge transfer.
>
> 2. The serial training paradigm in Eq. 3 is essential for effective and stable training. If trained directly in the form of the original Rectified flow, VP SDE and VE SDE, the student accuracy with WRN-40-2-WRN-16-2 pair on CIFAR-100 $\leq$ 5%. However, as demonstrated in Appendix M, the accuracy $\geq$ 72% if the FM-KD framework is used for training.
>
> 3. FM-KD is theoretically guaranteed and has been demonstrated to stable training with various meta-encoder and loss functions by a wide range of ablation experiments. By contrast, DiffKD is an empirical design with no theoretical guarantees. DiffKD prevents vanishing gradient through $L_{diffkd}$ outlined in its paper, but this form leads to unstable training.
>
> 4. We prove and explain the effectiveness of this training paradigm as it can approximate an implicit ensemble method in Proposition 3.2.
>
> 5. We design OFM-KD, which can be used in the Online KD scenario. Our proposed OFM-KD outperforms prior Online KD algorithms in most teacher-student pairs.
>
> 6. FM-KD is more effective than DiffKD on CIFAR-100 and ImageNet-1k, as shown in Table 1 & 2 and Appendix K & L of the revised paper. The experimental results in the original main paper where FM-KD did not reach SOTA performance were merely due to not utilizing the optimal settings.
>
> 7. FM-KD is more efficient than DiffKD, as demonstrated in Appendix J.
>
> 8. We design FM-KD$^\Theta$ in Sec. 3.5 of the revised paper in this discussion phase, which incurs no additional inference computational cost.
>
> 9. We unify VP ODE [1*,2*,3*], VE ODE [1*,2*,3*], and Rectified flow into the FM-KD framework in Appendix M in this discussion phase.

---

> ### Author Response · Authors · 2023-11-19
> **Response to Reviewer 6oRH (Part II)**
>
> ### **Q2. The FM-KD method proposed in this paper modifies the structure of the student network and increases the cost during model inference, which is a shortcoming of a model compression algorithm.**
>
> **A2.** Thank you for your valuable feedback. We address your concerns in the following three Parts.
>
> **Part 1:** FM-KD leads to significant performance improvements, and the added computational overhead in inference is acceptable. Additionally, FM-KD has the ability to adjust the Number of Function Evaluations (NFE) during inference to balance effectiveness and efficiency.
>
> **Part 2:** As demonstrated in Tables I and II, in comparison to DiffKD, a method that also elevates computational cost in inference, FM-KD is evidently more cost-effective regarding performance and GPU latency.
>
> Table I (ImageNet-1k)
>
> |Method | FM-KD | DiffKD |
> | ---- | ---- | ---- |
> | Meta-encoder | 2-MLP | 2-BottleNeck+Conv+BN+MLP |
> | Type | logit | both feature and logit |
> | R34-R18 | 73.17 | 72.49 |
> | R50-MBV1 | 74.22 | 73.78 |
>
> More details of Table I can be found in Appendix K of the revised paper.
>
> Table II (ImageNet-1k)
>
> | Method | DiffKD | FM-KD | DiffKD | FM-KD |
> | ---- | ---- | ---- | ---- | ---- |
> | Mode | training | training | inference | inference |
> | R50-MBV1 | 0.2431 | 0.1281 | 0.0517 | 0.0151 |
> | R34-R18 | 0.2376 | 0.1139 | 0.0390 | 0.0150 |
>
> Where x-MLP refers to x-layer MLP. The measurements in Table II were obtained under FM-KD using 2-MLP as meta-encoder. More details can be found in Appendix J of the revised paper.
>
> **Part 3:** We appreciate the opportunity to enhance our work and have designed FM-KD$^{\Theta}$ without extra computational costs in inference. For a comprehensive explanation, kindly refer to our detailed response regarding your Q4.
>
> ### **Q3. The computational cost during inference is not given in the paper.**
>
> **A3.** We have presented GPU latency measurements during inference in Table III. We can substantiate that the additional computational cost of FM-KD in inference is much lower compared with DiffKD.
>
> Table III (ImageNet-1k, hardware setting is 2*RTX 3080ti)
>
> | Method | DiffKD | FM-KD | FM-KD $^{\Theta}$| DiffKD | FM-KD | FM-KD $^{\Theta}$ | classical KD methods |
> | ---- | ---- | ---- | ---- | ---- | ---- | ---- | ---- |
> | Mode | training | training | training | inference | inference | inference | inference |
> | R50-MBV1 | 0.2431 | 0.1281 | 0.1281 | 0.0517 | 0.0151 | 0.0108 | 0.0108 |
> | R34-R18 | 0.2376 | 0.1139 | 0.1140 | 0.0390 | 0.0150 | 0.0108 | 0.0108 |
>
>
> ### **Q4. If refer to Fig. 5, the increase in cost is obvious and there is some unfairness in comparison with other methods.**
>
> **A4.** Thank you for your valuable comment.
>
> To ensure a fair comparison with other knowledge distillation algorithms such as DKD, DIST, and SPKD, we have further designed FM-KD$^{\Theta}$, which refines the process by distilling $Z_0$ obtained from FM-KD into an existing classification head (i.e. the original student's classification head) $\mathcal{T}_\mathrm{vanilla}(\cdot)$, thereby ensuring that no extra inference cost is incurred. More details can be found in Sec. 3.5 of the revised paper.
>
> Table IV (CIFAR-100)
>
> | Teacher | ResNet56 | WRN-40-2 | WRN-40-2 | ResNet32$\times$4 | VGG13 | VGG13 | WRN-40-2 |
> | ---- | ---- | ---- | ---- | ---- | ---- | ---- | ---- |
> | Student | ResNet20 | WRN-16-2 | WRN-40-1 | ResNet8$\times$4 | VGG8 | MobileNetV2 | ShuffleNetV1 |
> | Top-1 Acc. | 72.20 | 75.98 | 74.99 | 76.52 | 74.82 | 69.90 | 77.19 |
>
> Table V (ImageNet-1k)
>
> | T-S pair | R34-R18 | R50-MBV1 |
> | ---- | ---- | ---- |
> | Top-1 Acc. | 72.14 | 73.29 |
>
> The experimental results of FM-KD$^{\Theta}$ are presented in Tables IV and V, where the implementation details on ImageNet-1k and CIFAR-100 can be found in Appendix I. By comparing the accuracy of prior knowledge distillation algorithms like DKD, DIST, and SPKD in Tables 1 and 2 of the revised paper, we can conclude that FM-KD$^{\Theta}$ achieve SOTA performance across a wide range of teacher-student pairs.
>
> ### **Q5. Are there still significant advantages of the FM-KD approach over baseline models that increase the number of parameters to make time-consuming approximations?**
>
> **A5.** Thank you for your interesting question.
>
> Yes, DiffKD is an algorithm that increases the computational cost during inference. As indicated in Tables I and II, by utilizing a lightweight meta-encoder 2-MLP, FM-KD exceeds DiffKD, which uses 2-BottleNeck+Conv+BN+MLP as meta-encoder, by margins of 0.68% and 0.44% in the ResNet34-ResNet18 and ResNet50-MobileNetV2 pairs with a lower GPU latency (less than 1/3 DiffKD's in the ResNet50-MobileNetV2 pair), respectively.

---

> ### Author Response · Authors · 2023-11-19
> **Response to Reviewer 6oRH (Part III)**
>
> ### **Q6. The experimental performance is not outstanding enough. There is no significant advantage over other knowledge distillation methods in Table 3 of the experiments for object detection.**
>
> **A6.** Thank you for your feedback. We apologize for our inability to resolve this weakness in object detection. This result stems from the fact that in image classification, flow matching with the teacher at the logit level generally yields performance closer to that of the teacher. However, in object detection, flow matching with the teacher at the FPN (Feature Pyramid Network) level does not directly translate into improved performance in the final metric, mAP. We have already incorporated this point into our limitations (please see Sec. 7).
>
> ### **Q7. There are also several tests in the image classification experiments that do not perform as well as other existing methods.**
>
> **A7.** Thank you for your valuable comment. The experimental results of FM-KD in Tables 1 and 2 of the original paper are not obtained from optimal settings. It is demonstrated from Fig. 5 of the revised paper (Table 6 of the original paper) that FM-KD performs best on ImageNet-1k when using MLP as its architecture. Moreover, FM-KD can achieve better performance by using DKD as a loss function on CIFAR-100.
>
> We are pleased to experiment with the optimal settings, and the additional experimental results are presented in Tables VI and VII. It can be observed that FM-KD outperforms all prior methods and achieves SOTA performance.
>
> Table VI (ImageNet-1k, loss function is set as DIST, K=8)
>
> | T-S pair | Meta-encoder | R50-MBV1 |
> | ---- | ---- | ---- |
> | Top-1 Acc. (K=8) | 2-MLP | 74.22 |
> | Top-1 Acc. (K=8) | 3-MLP | 74.28 |
>
> Table VII (CIFAR-100, meta-encoder is set as Swin-Transformer, loss function is set as DKD)
>
> | T-S pair | VGG13-VGG8 | VGG13-MBV2 |
> | ---- | ---- | ---- |
> | Top-1 Acc. (K=1) | 75.21 | 69.68 |
> | Top-1 Acc. (K=2) | 74.86 | 69.52 |
> | Top-1 Acc. (K=4) | 75.42 | 69.94 |
> | Top-1 Acc. (K=8) | 74.46 | 69.94 |
>
> ### **References**
>
> [1*]. Score-Based Generative Modeling through Stochastic Differential Equations, ICLR 2021.
>
> [2*]. Flow Straight and Fast: Learning to Generate and Transfer Data with Rectified Flow, ICLR 2023.
>
> [3*]. Flow Matching for Generative Modeling, ICLR 2023.

---

### Official Review · Reviewer_J5cJ · 2023-11-01

**Soundness:** 3 good
**Presentation:** 2 fair
**Contribution:** 2 fair
**Rating:** 5
**Confidence:** 4

**Summary:**

This paper addresses knowledge distillation through improving feature representation matching from the student model to the pre-trained teacher model. Specifically, the authors extend an existing work DiffKD via replacing vanilla diffusion in DiffKD by Rectified Flow (another existing work which makes diffusion process as a flow having straight paths between any two steps). Experiments on image classification (with CIFAR-100 and ImageNet-1K datasets) and object detection (with MS COCO dataset) tasks are provided to show the effectiveness of the proposed method. Different baselines and setups are considered in experiments.

**Strengths:**

+  Improving feature representation matching is a critical problem in knowledge distillation research.

+ Leveraging diffusion process to augment feature representation matching for knowledge distillation is interesting, although the insight behind it is not very clear.

+ Image classification and object detection with large scale datasets like ImageNet-1K and MS COCO are considered for experimental comparison.

**Weaknesses:**

- The method and presentation.

This paper attempts to improve feature representation matching from knowledge distillation. The proposed method FM-KD heavily relies on two existing works DiffKD (arxiv 2023) and Rectified Flow (ICLR 2023). Specifically, the authors directly use Rectified Flow to replace vanilla diffusion in DiffKD, making the diffusion of student representation (either feature or logits) to have straight paths between any two steps. In general, FM-KD is rather incremental. The authors try to claim FM-KD as a totally new knowledge transfer framework, which is applicable to different types of teacher-student architectures, different metric-based distillation methods and different representations. However, it is mostly misleading, as DiffKD has already attained this goal when putting it in the context of the authors' viewpoint.

Furthermore, the underlying reason for why applying diffusion process to augment feature representation matching in knowledge distillation in not clear enough. What is the reasonable new technical insight here?

In addition, Rectified Flow assumes straight paths between any two steps during the diffusion, so is it reasonable when applying this design to feature representation matching in knowledge distillation? The original purpose of Rectified Flow is for faster diffusion but not more accurate performance. From Table 2 on ImageNet-1K, the proposed method FM-KD merely performs on par or worse than DiffKD.

- The limitations.

The authors did not discuss the limitations of the proposed method.

- The experiments.

As the proposed method is closely related to DiffKD, diffusion with Rectified Flow (straight paths) vs. diffusion with vanilla diffusion (non-straight paths), DiffKD and DiffKD+RectifiedFlow should be always used as the baselines instead of others.

From Table 2 on ImageNet-1K, the proposed method FM-KD merely performs on par or worse than DiffKD. This raises a critical problem, is it necessary to replace vanilla diffusion (non-straight paths, e.g. DDIM) by Rectified Flow (straight paths)? Why? On the other hand, Rectified Flow assumes straight paths for any two steps during the diffusion, is it reasonable to feature representation matching in knowledge distillation?

How about the extra cost to the inference with the trained student model? As the architecture of the trained student model is no longer same to the baseline, which violates the basic purpose of knowledge distillation, namely model compression.

- Others

There is no discussion on related works in the main paper. I noticed that the authors put this part in the Appendix, but this is not proper, to the best of my understanding.

**Questions:**

Please refer to my detailed comments in "Weaknesses" for details.

---

> ### Author Response · Authors · 2023-11-19
> **Response to Reviewer J5cJ (Part I)**
>
> Thank you for the thorough and constructive comments and suggestions. We think it may be helpful to clarify that probability flow and diffusion processes as generative models are of two different forms. Our research is specifically centered on Rectified flow, a component of probability flow. This flow-based generative model is built on the ODE forward form framework, setting it apart from diffusion processes that are commonly associated with the SDE forward form. A particularly noteworthy aspect of Rectified flow is its capacity to bridge two empirical distributions, a feature that diffusion processes are unable to achieve.
>
> Furthermore, we have carefully incorporated your comments in the revised paper. Please see our below responses to your questions and concerns one by one.
>
> ### **Q1. This paper attempts to improve feature representation matching from knowledge distillation. The proposed method FM-KD heavily relies on two existing works DiffKD (arxiv 2023) and Rectified Flow (ICLR 2023).**
>
> **A1.** Thank you for your valuable feedback. FM-KD draws some inspiration from Rectified flow, yet it is distinct from DiffKD. The primary motivation behind developing FM-KD was to tackle potential limitations we identified in the modeling approach used in DiffKD. Unlike DiffKD, which incorporates three extra empirical losses, FM-KD employs just a single additional loss from theoretical derivation and performs better than DiffKD in both effectiveness and efficiency (please see Tables 1 and 2 of the revised paper).
>
> **The main differences between FM-KD and DiffKD consist of three Parts:**
>
> **Part 1: The loss in FM-KD is different from the form of each loss in DiffKD.** The losses in DiffKD are redundant and stacked. DiffKD employs the original training loss of the diffusion model (i.e., $L_{diff}$) and combines it with two additional auxiliary losses (i.e., $L_{ae}$ and $L_{diffkd}$) as detailed in its original paper for training. By contrast, we utilize our proposed training loss $L_{FM-KD}$ (solely one), guaranteed by Theorem 3.1 in the original/revised paper, which differs from each of $L_{diff}$, $L_{ae}$, and $L_{diffkd}$. To the best of our knowledge, this serial training paradigm has not been previously introduced in existing work.
>
> **Part 2: FM-KD uses serial training paradigm while DiffKD uses original diffusion model training paradigm.** We believe the importance of our proposed serial training paradigm lies in its ability to effectively and stably propagate gradients back to the earlier layers of the student. By contrast, DiffKD might face challenges in functioning solely on the original diffusion model training paradigm, as it may struggle with preventing gradient vanishing. Though DiffKD aligns the sampling result $Z_0$ with the teacher feature/logit $X_T$ (i.e., $L_{diffkd}$ in DiffKD's original paper), ensuring backward gradient propagation to the student's earlier layers, DiffKD is highly sensitive to large Number of Function Evaluation (NFE) and the meta-encoder architecture in our experiments. We are pleased to note that FM-KD seems to avoid this particular issue.
>
> **Part 3: DiffKD=$X^S$->Gaussian noise->$X^T$ and FM-KD=$X^S$->$X^T$.** DiffKD does not immediately effectuate the transition from $X^S$ to $X^T$. Instead, it first transforms $X^S$ into a Gaussian noise and subsequently translates this noise to $X^T$. This dual-stage transformation process might potentially be overly complex. By contrast, FM-KD offers a more straightforward approach by directly forming a bridge between the distributions of student and teacher features/logits.
>
> We would like to note that the performance of DiffKD, which may be attributed to its heuristically design, could be perceived as not fully meeting the expected standards. For example, with the ResNet50-MobileNetV1 pair on ImageNet-1k, DiffKD obtains Top-1 Val Acc. 73.78% by increasing the GPU latency from 0.0108 to 0.0517 during inference, while FM-KD obtains Top-1 Val Acc. 74.22% by increasing the GPU latency from 0.0108 to 0.0151 during inference in Appendix J & K. Thus, FM-KD is more effective than DiffKD.

---

> ### Author Response · Authors · 2023-11-19
> **Response to Reviewer J5cJ (Part II)**
>
> ### **Q2. Specifically, the authors directly use Rectified Flow to replace vanilla diffusion in DiffKD, making the diffusion of student representation (either feature or logits) to have straight paths between any two steps. In general, FM-KD is rather incremental.**
>
> **A2.** Thank you for your insightful comment. In our work, we do not employ Rectified flow in its original form for knowledge distillation. There are significant modifications made from Eq. 2 (the original Rectified flow loss function) to Eq. 3 (our proposed serial loss function) in our paper. These enhancements are crucial. Without them, using the original Rectified flow training paradigm (as per Eq. 2) with the WRN-40-2-WRN-16-2 pair on CIFAR-100 would likely have led to the issue of vanishing gradients, resulting in a final test accuracy 3.42%.
>
> Additionally, it may be pertinent to highlight that FM-KD is not a mere incremental improvement. FM-KD represents a substantial deviation from existing training losses, such as those used in DiffKD. The training paradigm (i.e. Eq. 3) we introduced is not a mere extension but a distinct form. Unlike DiffKD, which relies on the original diffusion model training paradigm for training, FM-KD employs a uniquely formulated training paradigm, guaranteed with our proposed Theorem 3.1, to facilitate effective knowledge transfer. This design is quite critical. In our experiments, the original Rectified flow alone demonstrated insufficient results (e.g., achieving only 3.42% with the WRN-40-2-WRN-16-2 pair on CIFAR-100), but it can go up to 77.84% with our proposed unique serial training paradigm.
>
> ### **Q3. The authors try to claim FM-KD as a totally new knowledge transfer framework, which is applicable to different types of teacher-student architectures, different metric-based distillation methods and different representations. However, it is mostly misleading, as DiffKD has already attained this goal when putting it in the context of the authors' viewpoint.**
>
> **A3.** Although DiffKD has shown some progress towards being applicable across various teacher-student architectures and metric-based distillation methods, it has not effectively or adequately achieved it to the expected extent. We illustrate this in three Parts:
>
> **Part 1: FM-KD is less sensitive to architecture.** FM-KD exhibits good performance using various architectures as meta-encoder, as shown in our experiments with the ResNet50-MobileNetV2 pair (refer to Table 6 in the original paper or Fig. 5 in the revised paper). In contrast, DiffKD's performance appears more sensitive to architectural variations, experiencing training challenges such as loss becoming NAN under different meta-encoder configurations.
>
> **Part 2: FM-KD has evaluated this goal experimentally.** We have conducted comprehensive ablation studies to demonstrate that FM-KD exhibits generalization ability against variations in architectures and loss functions. On the other hand, it is observed that DiffKD, in its original paper, does not focus extensively on similar ablation studies.
>
> **Part 3: FM-KD can perform better.** FM-KD achieves better performance than DiffKD in both ResNet34-ResNet18 and ResNet50-MobileNetV1 pairs on ImageNet-1k with lower training and inference costs:
>
> Table I (ImageNet-1k)
>
> |Method | FM-KD | DiffKD |
> | ---- | ---- | ---- |
> | Meta-encoder | 2-MLP | 2-BottleNeck+Conv+BN+MLP|
> | Type | logit | both feature and logit |
> | R34-R18 | 73.17 | 72.49 |
> | R50-MBV1 | 74.22 | 73.78 |
>
> More details of Table I can be found in Appendix K of the revised paper.
>
> Table II (ImageNet-1k)
>
> | Method | DiffKD | FM-KD | DiffKD | FM-KD |
> | ---- | ---- | ---- | ---- | ---- |
> | Mode | training | training | inference | inference |
> | R50-MBV1 | 0.2431 | 0.1281 | 0.0517 | 0.0151 |
> | R34-R18 | 0.2376 | 0.1139 | 0.0390 | 0.0150 |
>
> Where x-MLP refers to x-layer MLP. The measurements in Table II were obtained under FM-KD using 2-MLP as meta-encoder. More details can be found in Appendix J of the revised paper.

---

> ### Author Response · Authors · 2023-11-19
> **Response to Reviewer J5cJ (Part III)**
>
> ### **Q4. Furthermore, the underlying reason for why applying diffusion process to augment feature representation matching in knowledge distillation in not clear enough. What is the reasonable new technical insight here?**
>
> **A4.** It may be helpful to clarify that our method does not employ a diffusion process, typically associated with SDE forward form. Instead, our approach is grounded in Rectified flow, which belongs to ODE forward form. The motivation for applying Rectified flow to augment feature representation matching (i.e. knowledge transfer) is as follows
>
> 1. Rectified flow can establish a bridge between two empirical distributions. This characteristic makes the Rectified flow particularly suitable for knowledge distillation in transferring knowledge from the teacher to the student, as both teacher and student features/logits are sampled from empirical distributions.
>
> 2. It is difficult to accomplish the knowledge transfer between two complex empirical distributions by one step. Thus, adopting the multi-step sampling from Rectified flow can realize the progressive transformation (i.e. probability path) from the student feature/logit distribution to the teacher feature/logit distribution, which leads to effective knowledge transfer.
>
> 3. As proven in Proposition 3.2 of our original/revised paper, FM-KD is implemented through Rectified flow and can be approximated as an implicit ensemble method, which is why FM-KD can achieve more precise knowledge transfer. Unfortunately, other noise schedules, such as VP ODE [*1,*3,*4] and VE ODE [*1,*3,*4], cannot achieve this goal.
>
> 4. FM-KD has an property that can be used to realize the trade-off between performance and efficiency by changing NFE in inference.
>
>
> ### **Q5. In addition, Rectified Flow assumes straight paths between any two steps during the diffusion, so is it reasonable when applying this design to feature representation matching in knowledge distillation? The original purpose of Rectified Flow is for faster diffusion but not more accurate performance.**
>
> **A5.** Thank you for your insightful question, our response consists of three points.
>
> 1. In our theoretical derivation, Rectified flow can be approximated an implicit ensemble method for effective knowledge transfer, other noise schedules, such as VP SDE [1*,3*,4*] and VE SDE [1*,3*,4*] cannot. This contrasts with other noise schedules like VP SDE and VE SDE, which may not be as effective due to the varying influence of truncation errors in non-straight noise schedules on the final result $Z_0$ from the Euler method.
>
> 2. Rectified flow can accelerate sampling (equivalent to minimizing the curvature) [2*] making it possible to be effective with very small NFE scenarios. Consequently, the fewer the NFEs, the smaller the training and inference computational burden, so we apply Rectified flow for feature representation matching.
>
> 3. Rectified flow is not the only implementation of FM-KD. The assumption that the trajectory between two empirical distributions is straight is merely one modeling approach in optimal transport theory. It's viable, but not the only one. We unify VP ODE (w.r.t. VP SDE), VE ODE (w.r.t. VE SDE) and Rectified flow in FM-KD and present this in Appendix M of the revised paper. Interestingly, we discover that Rectified flow performed best, perhaps because Rectified flow can be approximated an implicit ensemble method.
>
> ### **Q6. From Table 2 on ImageNet-1K, the proposed method FM-KD merely performs on par or worse than DiffKD.**
>
> **A6.** The experimental results (i.e. Table 2 of the original paper) for ImageNet-1k are not optimal for FM-KD. The results in Tables 1 and 2 of the original main paper are presented only to ensure consistency of the loss function (i.e. DIST) and the meta-encoder (i.e. Swin-Transformer). As illustrated in Fig. 5 of the revised paper (Table 6 of the original paper) that FM-KD performs best on ImageNet-1k when using MLP as its architecture. Therefore, we are pleased to provide the best performance of FM-KD on ImageNet-1k by using MLP as meta-encoder:
>
> Table III (ImageNet-1k)
>
> |Method | FM-KD | FM-KD | FM-KD | DiffKD |
> | ---- | ---- | ---- | ---- | ---- |
> | Meta-encoder | 1-MLP | 2-MLP | 3-MLP | 2-BottleNeck+Conv+BN+MLP |
> | Type | logit | logit | logit | both feature and logit |
> | R34-R18 | 72.48 | 73.17 | 73.28 | 72.49 |
> | R50-MBV1 | 73.74 | 74.22 | 74.28 | 73.78 |
>
> More details of Table III can be found in Appendix K of the revised paper.
>
>
> ### **Q7. The authors did not discuss the limitations of the proposed method.**
>
> **A7.** Sorry that our oversight caused you concern, we have added limitation to Sec. 7 of the revised paper.

---

> ### Author Response · Authors · 2023-11-19
> **Response to Reviewer J5cJ (Part IV)**
>
> ### **Q8. As the proposed method is closely related to DiffKD, diffusion with Rectified Flow (straight paths) vs. diffusion with vanilla diffusion (non-straight paths), DiffKD and DiffKD+RectifiedFlow should be always used as the baselines instead of others.**
>
> **A8.** Thanks for this suggestion, and here we provide more experiments and discussions.
>
> **For diffusion with vanilla diffusion (non-straight paths)**, we clarify that it is not feasible to compare FM-KD with vanilla diffusion processes since vanilla diffusion processes simply do not work in knowledge transfer. Specifically, in our exploratory studies, we try most of the ODE form of popular diffusion processes (non-straight paths), including VP ODE (converted from VP SDE and DDPM) and VE ODE (converted from VE SDE and NCSN). We also try vanilla Rectified flow (straight path). However, if these methods use the vanilla training paradigm:
>
> Score function matching in VP ODE and VE ODE:
>
> $ argmin_{v_\theta} E_{(Z_1,Z_0,t)} ||s_{v_\theta}(Z_t,t) - \nabla_{Z_t} \log p(Z_t|Z_0)||_2^2.  $
>
> Velocity matching in Rectified flow:
>
> $ argmin_{v_\theta} E_{(Z_1,Z_0,t)}||g_{v_\theta}(Z_t,t) - \frac{\partial Z_t}{\partial t}||_2^2. $
>
> In our experiments, their final test accuracy will not exceed 5% with WRN-40-2-WRN-16-2 pair on CIFAR-100 (suffer from vanishing gradient). After we introduce the training paradigm in Eq. 3, as guaranteed by Theorem 3.1, Rectified flow can finally work effectively. Thus, the central focus of our paper is the training paradigm presented in Eq. 3. More importantly, we unify VP ODE, VE ODE, and Rectified flow within FM-KD through Theorem 3.1 in Appendix M, and we experimental demonstrate the effective modeling of both VP ODE and VE ODE. However, Rectified flow outperforms VP ODE and VE ODE in terms of performance and stability.
>
> **For DiffKD**, we have presented additional comparative experimental results in Table III and substantiated that FM-KD is more effective than DiffKD. Furthermore, we conduct DiffKD to use the meta-encoder architecture (i.e. 2-MLP) from FM-KD, and FM-KD to use the meta-encoder architecture (2-Bottleneck+Conv+BN+MLP) from DiffKD, with the results presented in Table IV. The final results indicate that "DiffKD uses FM-KD's meta-encoder" underwent training collapse at epoch 1, whereas FM-KD (i.e. 74.26%) significantly surpassed DiffKD (i.e. 73.78%) in test accuracy when using the same meta-encoder architecture as DiffKD.
>
> Table IV (with ResNet50-MobileNetV1 pair on ImageNet-1k)
>
> | Method |DiffKD uses FM-KD's meta-encoder | FM-KD uses DiffKD's meta-encoder | DiffKD uses DiffKD's meta-encoder | FM-KD uses FM-KD's meta-encoder |
> | ---- | ---- | ---- | ---- | ---- |
> Top-1 Acc. | NAN | 74.26 | 73.78 | 74.22 |
>
> **For DiffKD+RectifiedFlow**, we implemented subtle adjustments to the parameters $\alpha_t$ and $\sigma_t$ in $p(Z_t|Z_0) = \mathcal{N}(Z_t | \alpha_tZ_0,\sigma_t^2I)$, setting them to $1-t$ and $t$, respectively. These modifications were accompanied by a learning rate warm-up strategy specifically for DiffKD+Rectified flow applied to CIFAR-100. The results of these experiments are presented in Table V. From these results, it appears that FM-KD tends to be more effective in comparison to DiffKD+Rectified flow.
>
> Table V (CIFAR-100)
>
> | Teacher | ResNet56 | WRN-40-2 | WRN-40-2 |
> | ---- | ---- | ---- | ---- |
> | Student | ResNet20 | WRN-40-1 | WRN-16-2 |
> | DiffKD+Rectified flow | 72.00 | 74.28 | 76.07 |
> | DiffKD | 71.92 | 74.09 | 76.13 |
> | FM-KD (K=4) | 75.12 | 76.24 | 77.69 |
>
>
> ### **Q9. From Table 2 on ImageNet-1K, the proposed method FM-KD merely performs on par or worse than DiffKD. This raises a critical problem, is it necessary to replace vanilla diffusion (non-straight paths, e.g. DDIM) by Rectified Flow (straight paths)? Why?**
>
> **A9.** We address your concerns in the following two points.
>
> 1. The experimental results of FM-KD (i.e. Table 2 of the original paper) on ImageNet-1k are not obtained from optimal settings. We conduct additional experiments and the additional experimental results in Table III demonstrate that FM-KD substantially outperforms DiffKD.
>
> 2. DDIM cannot be used to replace Rectified flow since DDIM is merely an ODE solver and is not related to whether 'the path is straight or not'. We think you are referring to different ODE processes with non-straight paths, such as VP ODE (converted from VP SDE and DDPM) and VE ODE (converted from VE SDE and NCSN). In Appendix M of our revised paper, we unify VP ODE, VE ODE, and Rectified flow into the FM-KD framework for efficient training. The experimental results in Appendix M demonstrate that training with Rectified flow is more effective and stable than using VP ODE and VE ODE in the WRN-40-2-WRN-16-2 pair on CIFAR-100.

---

> ### Author Response · Authors · 2023-11-19
> **Response to Reviewer J5cJ (Part V)**
>
> ### **Q10. How about the extra cost to the inference with the trained student model? As the architecture of the trained student model is no longer same to the baseline, which violates the basic purpose of knowledge distillation, namely model compression.**
>
> **A10.** We address this review in two Parts:
>
> **Part 1:** We have presented GPU latency measurements during inference in Table VI. We can substantiate that the additional computational cost of FM-KD in inference is much lower compared with DiffKD.
>
> Table VI (ImageNet-1k, hardware setting is 2*RTX 3080ti)
>
> | Method | DiffKD | FM-KD | FM-KD $^{\Theta}$| DiffKD | FM-KD | FM-KD $^{\Theta}$ | classical KD methods |
> | ---- | ---- | ---- | ---- | ---- | ---- | ---- | ---- |
> | Mode | training | training | training | inference | inference | inference | inference |
> | R50-MBV1 | 0.2431 | 0.1281 | 0.1281 | 0.0517 | 0.0151 | 0.0108 | 0.0108 |
> | R34-R18 | 0.2376 | 0.1139 | 0.1140 | 0.0390 | 0.0150 | 0.0108 | 0.0108 |
>
>
> **Part 2:** To ensure a fair comparison with other knowledge distillation algorithms such as DKD, DIST, and SPKD, we have further designed FM-KD$^{\Theta}$, which refines the process by distilling $Z_0$ in FM-KD into an existing classification head (i.e. the original student's classification head) $\mathcal{T}_\mathrm{vanilla}(\cdot)$, thereby ensuring that no extra inference cost is incurred. More details can be found in Sec. 3.5 of the revised paper.
>
> Table VII (CIFAR-100)
>
> | Teacher | ResNet56 | WRN-40-2 | WRN-40-2 | ResNet32$\times$4 | VGG13 | VGG13 | WRN-40-2 |
> | ---- | ---- | ---- | ---- | ---- | ---- | ---- | ---- |
> | Student | ResNet20 | WRN-16-2 | WRN-40-1 | ResNet8$\times$4 | VGG8 | MobileNetV2 | ShuffleNetV1 |
> | Top-1 Acc. | 72.20 | 75.98 | 74.99 | 76.52 | 74.82 | 69.90 | 77.19 |
>
> Table VIII (ImageNet-1k)
>
> | T-S pair | R34-R18 | R50-MBV1 |
> | ---- | ---- | ---- |
> | Top-1 Acc. | 72.14 | 73.29 |
>
> The experimental results of FM-KD$^{\Theta}$ are presented in Tables VII and VIII, where the implementation details on ImageNet-1k and CIFAR-100 can be found in Appendix I of our revised paper. By comparing FM-KD$^{\Theta}$ with prior knowledge distillation algorithms like DKD, DIST, and SPKD in Tables 1 and 2 of the revised paper, we can conclude that FM-KD$^{\Theta}$ achieve SOTA performance across a wide range of teacher-student pairs.
>
> ### **Q11. There is no discussion on related works in the main paper. I noticed that the authors put this part in the Appendix, but this is not proper, to the best of my understanding.**
>
> **A11.** Thank you for pointing out this presenting problem in our paper. We have moved Related work to Sec. 5 in the main paper.
>
> ### **References**
>
> [1*]. Score-Based Generative Modeling through Stochastic Differential Equations, ICLR 2021.
>
> [2*]. Minimizing Trajectory Curvature of ODE-based Generative Models, ICML 2023.
>
> [3*]. Flow Straight and Fast: Learning to Generate and Transfer Data with Rectified Flow, ICLR 2023.
>
> [4*]. Flow Matching for Generative Modeling, ICLR 2023.

---

### Official Review · Reviewer_GAWr · 2023-12-01

**Soundness:** 2 fair
**Presentation:** 2 fair
**Contribution:** 1 poor
**Rating:** 3
**Confidence:** 5

**Summary:**

The paper proposes diffusion-like distillation loss named FM-KD. In FM-KD, a meta-encoder converts student representation similar to teacher representation with a recursive process. FM-KD uses all representations during the recursive conversion for KD loss. FM-KD can be expanded to an online KD called OFM-KD. Experiments show the improvement of FM-KD in KD benchmark.

**Strengths:**

- Introducing a meta-encoder for inference is an interesting approach.

**Weaknesses:**

- Performance improvement in offline-KD is marginal.
  - In Table 1, FM-KD$^\Theta$ shows marginal improvements (<0.3%p) compared to ReviewKD and DKD.
  - In Table 2, FM-KD$^\Theta$ shows a similar performance to DIST.
  - In Table 3, the improvement of FM-KD compared to PKD is small (<0.2)
- The meta-encoder increases computation costs for inference, which is not acceptable in KD benchmark.
  - I think it is unfair to compare FM-KD (K=?) with other distillation such as ReviewKD and DKD which doesn't use additional modules at inference.
  - The paper lacks analysis and reports on the amount of additional computation. Inference costs for each network should be reported for every K value. Figure 9 is not enough.
  - There are effective ways to improve network performance with additional computation, like SE module. FM-KD (K=?) should be compared with them to claim an effective way to increase performance with additional computation.
- Theoretical proof doesn't look valuable
  - Appendix B shows that the recursion will make students similar to the teacher. I don't understand how it is connected to distillation performance and gradient vanishing in an early layer.
  - Appendix C shows that FM-KD approximates the ensemble. However, it is possible to use an ensemble in distillation like ONE [A] and multi-exit [B]. Especially, FM-KD is similar to multi-exit [B] made by additional modules. Thus, it has limited novelty.
    - [A] Knowledge Distillation by On-the-Fly Native Ensemble, NIPS 2018
    - [B] Distillation-Based Training for Multi-Exit Architectures, ICCV 2019
- Experiments are significantly out-dated.
  - The paper uses traditional distillation benchmarks in Tables 1,2,4 and 5. These benchmarks are based on old baselines with small models, which makes it hard to contribute to recent training trends. I recommend authors apply FM-KD to recent training recipes such as ViT and Swin to enhance the impacts of paper for the general field.

**Questions:**

.

---

### Author Response · Authors · 2023-11-19
**Overview of Revisions**

Thank all reviewers for their valuable comments and suggestions. We have made the following revisions to the paper:

### **For method:**

1. We have introduced the lightweight FM-KD$^{\Theta}$ in Sec. 3.5 to address the potential increase in computational burden during inference associated with FM-KD and present the experimental results in Tables 1 and 2 (**Reviewer J5cJ and 6oRH**).

2. We unify VP SDE, VE SDE, and Rectified flow in FM-KD and experimentally demonstrate that Rectified flow is the most stable and efficient choice in Appendix M (**Reviewer J5cJ and 6oRH**).

### **For experiment:**

1. We have updated these results in the revised paper, ensuring that FM-KD now achieves SOTA performance across all teacher-student pairs (**Reviewer J5cJ, 6oRH and c4jx**).

2. We have added GPU latency measurements on FM-KD, FM-KD$^{\Theta}$ and DiffKD during training and inference in Appendix J (**Reviewer J5cJ and 6oRH**).

3. We have added additional ablation and comparison studies on ImageNet-1k to substantiate that FM-KD is more effective than DiffKD in Appendix K (**Reviewer J5cJ, 6oRH and c4jx**).

4. We have performed architecture-sensitive experiments with FM-KD and DiffKD in Appendix L (**Reviewer J5cJ and 6oRH**).

### **For presentation:**

1. We have moved Related work from Appendix to Sec. 5 of our revised main paper (**Reviewer J5cJ**).

2. We have added limitation in Sec. 7 (**Reviewer J5cJ**).

3. To save space in our main paper, we have moved Implementation details to Appendix I and adjusted Table 6 of the original paper to Fig. 5 of the revised paper.

4. We have added additional descriptions of the differences between FM-KD and DiffKD.

---

### Meta-Review · Area_Chair_QAxM · 2023-12-02

**Metareview:**

Reviewers found the paper to have an interesting perspective on knowledge distillation, but raised a number of concerns around the novelty compared to DiffKD, significance of empirical results, comparison to baselines, additional computational cost of the method, and value of the theoretical contributions. For the last point, it appears that the theoretical results are not currently stated in a sufficiently precise manner (Theorem 3.1 and Proposition 3.2 should state precise mathematical statements that will be proven; it is not clear how a proof of Proposition 3.2 can in two places make use of an $\approx$ without discussion of the residual in the approximation).

The author response included new results to address several of these points, which is greatly appreciated. However, it would be appropriate for the large number of new results to undergo a fresh round of review. Further, there remain concerns around the theory.

**Justification For Why Not Higher Score:**

Concerns around the novelty compared to DiffKD, significance of empirical results, comparison to baselines, additional computational cost of the method, and value of the theoretical contributions.

**Justification For Why Not Lower Score:**

N/A

---

### Decision · Program_Chairs · 2024-01-16

Reject